# The Cost of Balanced Training-Data Production in an Online Data Market

## ABSTRACT

Many ethical issues in machine learning are connected to the training data. Online data markets are an important source of training data, facilitating both production and distribution. Recently, a trend has emerged of for-profit "ethical" participants in online data markets. This trend raises a fascinating question: Can online data markets sustainably and efficiently address ethical issues in the broader machine-learning economy?

In this work, we study this question in a stylized model of an online data market. We investigate the effects of intervening in the data market to achieve balanced training-data production. The model reveals the crucial role of market conditions. In small and emerging markets, an intervention can *drive the data producers out of the market*, so that the cost of fairness is maximal. Yet, in large and established markets, *the cost of fairness can vanish* (as a fraction of overall welfare) as the market grows.

Our results suggest that "ethical" online data markets can be economically feasible under favorable market conditions, and motivate more models to consider the role of data production and distribution in mediating the impacts of ethical interventions.

ACM Reference Format:
Anonymous Author(s). 2018. The Cost of Balanced Training-Data Production in an Online Data Market. In *Proceedings of Make sure to enter the correct conference title from your rights confirmation emai (Conference acronym 'XX).* ACM, New York, NY, USA, 22 pages. https://doi.org/XXXXXXX.XXXXXXX

## 1 INTRODUCTION

It is now widely recognized that machine-learning systems can raise ethical issues, including issues of fairness, privacy, law, and working conditions. Models can produce predictions that systematically differ by race [40], reproduce the photos of individuals in their training data [11], be trained on copyrighted materials [37], and require the labeling of psychologically harmful content [44]. Ethical issues have been documented across diverse applications and domains [5, 8, 10].

Much work on addressing these issues has focused on the training data. Researchers have developed methods to analyze, transform, or augment a given dataset [14, 22, 54], and built new training-data resources [8, 39, 53]. Non-profit organizations have supported the production of training data through innovation and subsidization [23, 36]. Governments have reviewed and improved their data infrastructure, emphasizing data that is widely used, including for

machine learning [41, 42]. But these efforts have been largely restricted to academia, non-profit organizations, and government.

Recently, a small and growing number of firms have been attempting to address ethical issues in training data as profit-seeking participants in online data markets. They are targeting a range of issues including protecting creators' copyright [50], creating job opportunities with fair working conditions [29], and producing representative training data [18]. And they are working together [4]. This trend is intriguing.

On the one hand, online data markets offer powerful possibilities. Ethical issues in the training data could be addressed right at the source—at the level of training-data production. Online infrastructure enables monetization and transaction on a web scale, and can facilitate greater participation by under-represented and disadvantaged communities [15]. On the other hand, online data markets face economic pressures that may render addressing ethical issues in a free-market environment economically unviable. If an ethical intervention increases production costs, this may diminish the economic surplus of machine learning, or place a firm at a disadvantage to less scrupulous competitors. This fledgling trend raises an exciting question: *Can online data markets sustainably and efficiently address ethical issues in the machine-learning economy?*

In this work, we take a first step towards answering this question. We study the problem of *unbalanced training data*, loosely defined here (see Section (3.5) for a precise definition) as there being too few samples of some group (such as racial group or gender) in the training data. Unbalanced training data is linked to a number of issues in fair machine learning including performance disparities and equity assessments [3, 8, 14, 25]. We investigate the effects of intervening to achieve balanced training-data production in an online data market. Our aim is to motivate further work by shedding light on the economics of the intervention: *How costly can it be to achieve balanced training-data production?*

An intervention changes the training-data demographics, but also constrains the extraction of economic value. If fewer revenues are extracted, then the budget for data production will decrease and sellers will produce fewer training samples. In contrast to works that assume a fixed budget for data production as in [9, 21], the cost of fairness, loosely defined here as the loss in efficiency (see Section (3.5) for details), now arises out of a complex dynamic that includes this feedback and it is unclear what the overall impact will be. *How does the cost of fairness behave?*

*Summary of contributions.* We investigate this question in a stylized model of an online data market. Our main contributions are as follow.

- We revisit a well-known model of Agarwal et al. [2]. Our modeling contribution is to formulate a specialized variant that endogenizes data production and allows the marketplace to impose a fairness constraint. Our model captures two important factors that can lead to unbalanced

training-data production absent an intervention. First, data production is endogenous: sellers decide *how many training samples to produce for each group* in order to maximize their profit. Second, groups can systematically differ on three dimensions that influence sellers' profits: 1) the potential economic value that can be extracted; 2) the difficulty of the prediction tasks; and 3) the training-sample production costs.

- We describe the Nash equilibria of the model in a tractable and restricted *quasi-symmetric setting*. Our results reveal the equilibrium outcomes in a baseline scenario, when no fairness intervention is undertaken, and in an intervention scenario, when the marketplace undertakes a fairness intervention.

- We highlight the effects of applying fairness interventions in small and emerging markets. We characterize how imposing a fairness intervention from the get-go interacts with market formation. When the potential economic value is small for all groups, *the intervention can backfire*. The cost of fairness can be unbearable and the market may not form at all. Thus, every agent loses the full utility it would have received without an intervention. Worse still, this can harm the very groups the intervention is intended to benefit—leading to decreased accuracy and data produced for them.

- Beyond emerging markets, we consider large and established markets, where we find that the the cost of fairness can be completely offset. When the potential economic value *of at least one group* is sufficiently large and growing, the intervention can affect agents in only two positive ways. Some agents can be strictly better off; the intervention can have a positive externality that benefits some market participants in addition to creating more data and higher accuracy for the intended groups. But perhaps more surprisingly, for *all* the other market participants—the sellers, the buyers, and the marketplace—*the cost of fairness amortizes*, i.e., the cost of fairness, as a fraction of their utilities without intervention decreases, even vanishes, and becomes arbitrarily small.

*Paper Organization.* The remainder of this paper is organized as follows. We discuss related work in Section (2). We present our model in Section (3). We analyze the model equilibria in Section (4). In Section (5), we study the conditions under which a fairness intervention can backfire in the market. We study the conditions under which the cost of fairness amortizes in Section (6). We conclude with some limitations and discussion in Section (7)

## 2 RELATED LITERATURE

We study the cost of fairness in an online data market. The cost of fairness has been studied in a large literature. We mention three important threads: 1) characterizing the behavior of the cost of fairness for a given fairness criterion [16, 35]; 2) working with the cost of fairness by formulating relaxations or novel variants of fairness criteria [13, 26]; and 3) studying the impact of interacting fairness criteria on the cost of fairness [19, 31]. Our contribution is to show that the *relative severity* of the cost of fairness can vary, in

particular, that it can be amortized by economic growth. Our hope is that this may spin off a new thread in this literature.

We formulate our model to capture, as special cases, a wide range of real-world online data markets. The literature on real-world online data markets has focused on issues such as the structure of data markets [46, 48, 49], pricing mechanisms [6, 47], and challenges to market transactions [12, 30]. To the best of our knowledge, the recent trend of emerging ethical data markets has not yet received research attention. Ours is the first theoretical work to try and understand this trend and its potential.

Although our focus is on fairness, there is a significant theoretical literature on privacy and data markets. This literature has studied how to reconcile privacy and efficiency through pricing mechanisms and architectural blueprints [24, 32, 33]. Our work shows initial promise for reconciling fairness and efficiency. This suggests that it may be possible to reconcile the efficiency of data markets with ethical issues more broadly.

## 3 ONLINE DATA MARKET MODEL

We design our model to capture a wide range of real-world and theoretical online data markets. We take the theoretical model of Agarwal et al. [2] as a starting point. Their model is a practical blueprint for an end-to-end automated, computationally efficient, and real-time online data market for buying machine-learning predictions and selling training data. We formulate a specialized variant that endogenizes training-data production and enables the marketplace to impose a fairness constraint.

Formally, our data market is comprised of three kinds of agents: 1) $M$ sellers that produce and sell data to the marketplace; 2) $N$ buyers who buy predictions from the marketplace; and 3) a centralized marketplace that coordinates the market by aggregating and allocating data, producing predictions by carrying out machine learning, and setting prices. We next define each kind of agent and their role in the data market.

### 3.1 Datasets and sellers

Our model endogenizes data production: sellers respond to market conditions by deciding whether and what data to produce.

*Datasets* are a widespread form of data supply in real-world and theoretical online data markets [9, 21]. Datasets are typically organized into discrete elements called samples. To model fairness, we also suppose that each sample is exclusively associated to some group. A dataset could could be comprised of images of people, each sample could be a single image of a person, and the group could be the gender, age, or race of that person.

**Definition 3.1.** *(Dataset) A dataset is made up of samples; each sample is exclusively associated to one group $g$, from a possible set of groups $G$. The dataset is described by a vector $x \in \mathbb{R}^{|G|}$ where $x_g \geq 0$ gives the number of samples associated to group $g$ in the dataset $x$. We denote the total number of samples in $x$, by $\|x\| \triangleq \sum_{g \in G} x_g$.*

*Sellers* produce datasets. Each seller decides how many samples to produce of each group. Bearing on their decision are production costs. We use a simple model of production costs where each sample costs a fixed amount. This is similar to sample-based pricing in real-world data markets [43]. A common model in the fair machine learning literature extends this to group-specific costs [9, 21].

For example, a seller may produce text data including text from different languages where the languages may be considered different groups. Each word in a widely-used language may cost \$0.001 to produce, whereas each word in a rarely-used language may cost \$0.003 to produce.

**Definition 3.2.** *(Sellers) There are M sellers. Each seller j produces a dataset $x^{(j)}$ to sell. We assume that, for each group g, seller j's production process ensures that the $x_g^{(j)}$ samples in the dataset are independent and identically distributed, and that samples are drawn independently between groups.*

*Each seller j faces a production-cost structure, $\kappa^{(j)} \in \mathbb{R}^{|G|}; \kappa_g^{(j)} > 0$ is the constant marginal cost incurred by seller j to produce a sample of group g. The cost to seller j of producing dataset $x^{(j)}$ is $\sum \kappa_g^{(j)} x_g^{(j)}$, or in vector notation, $\kappa^{(j)T} x^{(j)}$.*

## 3.2 Data demand and the buyers

Demand for training data is driven by valuable machine-learning applications. We decompose the value of a machine-learning application into a machine-learning component, *prediction task*, and an economic component, *value of accuracy*.

A *prediction task* distills an application problem into a form that is workable by machine learning. For example, transcribe the words spoken in a speech clip, score the risk of a borrower defaulting on a loan, or conjecture a person's gender from an image of their face.

We model a prediction task in terms of its *learning curve*, $\mathcal{G}(\cdot)$ which captures the relationship between the training data, $x$, and accuracy, $\mathcal{G}(x)$. Learning curves are usually modeled as a function of the number of samples in the training data [27, 28, 52]. But in our model, a dataset $x$ is a vector that also captures the dataset demographics. Prior work shows that dataset demographics affect accuracy [3, 25], so presumably it affects the learning curve.

In general, it remains unclear exactly how to model learning curves as a function of both the number of training samples and their demographics, and assumptions must be made. One can expect that more training samples always help, and that a sample typically contributes more to accuracy for prediction on similar demographic groups. While we believe that this is an important issue to consider in future work, we make the following simplifying assumptions that still allows us to focus here on the economics of fairness intervention: (1) We analyze the scenario in which the value is extracted by buyers separately among groups, so the prediction tasks may be written as group-specific. This captures situations in which group-specific models are used or group-aware models are allowed and preferred. For example, the state of Wisconsin ruled that considering gender is permissible in recidivism prediction because it increases accuracy [45]. (2) Only the training samples of the associated group contribute to learning on its prediction task.

**Assumption 3.1.** *(Zero inter-group transfer) Let x be a dataset, and $\mathcal{G}$ be a prediction task associated to some group g, then*

$$\mathcal{G}(x) = \mathcal{G}(x_g). \tag{1}$$

*Note that we are abusing notation here, x is a vector whereas $x_g$ is a scalar, but we hope that this clarifies that when the whole dataset is passed, only the training samples of the group g are informative.*

To be clear, prediction tasks for different group can differ or follow the same exact learning curve. Here we only assume that they are not interacting through transfer learning. Assumption (3.1) may be seen as a simplified extreme version of the assumption, common in the fair machine learning literature, that machine learning usually does not fully transfer across populations [16, 20]. Note that, intuitively, some transfer learning ought to reduce learning disparities among group. One can therefore informally interpret that this model overestimates the cost of fairness. This provides some indication that our results on amortizing the cost of fairness (see Section (6)) are robust.

By Assumption (3.1), a learning curve depends only on the number of samples of an associated group. We use a common learning-curve model [21, 27, 28, 52] that captures three of their fundamental properties: 1) accuracy increase as the amount of training data increases; 2) the gain in accuracy is diminishing; and 3) there is a limit to the maximum possible accuracy.

**Definition 3.3.** *Learning curve A learning curve, $\mathcal{G}$, is defined by three parameters: Z, $\alpha$, and $\beta$. Given n training samples, the accuracy, $\mathcal{G}(n)$, is defined to be*

$$\mathcal{G}(n) \triangleq \left( Z - \alpha n^{-\beta} \right)_+, \tag{2}$$

*where $(\cdot)_+$ denotes the positive part.*

We model the economic component of data demand by a *marginal value-of-accuracy*, $\mu$, that we assume is constant. Thus, the value of $x_g$ training samples for a prediction task associated to group $g$ and with a learning curve $\mathcal{G}$ is $\mu\mathcal{G}(x_g)$. This is the amount that a buyer with prediction task[1] $\mathcal{G}$ is willing to pay for predictions derived from $x_g$ samples. For example, an ecommerce company may estimate that every 1% of accuracy in transcribing spoken words yields revenues of \$10,000. If $x_g$ training samples results in overall accuracy of 73%, the company would expect revenues of \$730,000 and be willing to pay up to that amount for those predictions.

**Definition 3.4.** *(Buyers) There are N buyers. Each buyer i faces |G| prediction tasks, one for each group g. Each prediction task has its own learning curve with parameters $Z_{i,g}$, $\alpha_{i,g}$, and $\beta_{i,g}$. Buyer i has a private value-of-accuracy $\mu_{i,g}$ for each group-specific prediction task; we collectively denote these $\mu_i$.*

## 3.3 Market mechanics and the marketplace

The marketplace coordinates the market. The sellers give the marketplace access to their datasets and the buyers disclose to the marketplace their values-of-accuracy and prediction tasks. The role of the marketplace is to perform three functions: 1) produce predictions; 2) set prices; and 3) divide payment between the sellers.

*Market Mechanics.* Market transactions begin with an interaction between the sellers and the marketplace. Each seller $j$ gives the marketplace access to its dataset, $x^{(j)}$. The marketplace can combine the sellers' datasets for machine learning and revenue division. For a subset of sellers $T \subseteq [M]$, the dataset $x^{(T)}$ is defined

---

[1]Since a prediction task defines a learning curve, we refer to $\mathcal{G}$ as either depending on which makes more sense in the context.

coordinate-wise by,[2]

$$x_g^{(T)} = \sum_{j \in T} x_g^{(j)}. \tag{3}$$

In particular, $x^{([M])}$ is the aggregate dataset of all the sellers' datasets. With access to the sellers' data, the marketplace has one of the key production factors for predictions.

Next, the marketplace chooses a reserve-price vector, $p \in \mathbb{R}^{|G|}$, $p_g > 0$, and then publicly announces $p$ so that buyers know $p$ at the beginning of the next step.

Once $p$ is announced, each buyer $i$ submits its prediction tasks and a bid vector $b_i \in \mathbb{R}^{|G|}$ to the marketplace. The bid vector $b_i$ is buyer $i$'s report to the marketplace of its values-of-accuracy $\mu_i$.

Then, the marketplace allocates training samples from the aggregate dataset to carry out machine learning and produce predictions for the buyers' prediction tasks. The marketplace uses a reserve price allocation mechanism, $\mathcal{AF}_g$, for each group $g$. For each group $g$, the marketplace allocates all the samples in the aggregate dataset to the machine learning for buyer $i$'s prediction task for group $g$ if the buyer bids at least the reserve price $p_g$, and otherwise nothing. Formally,

$$\mathcal{AF}_g(b_{i,g}, x^{([M])}) \triangleq \begin{cases} x_g^{([M])} & \text{if } b_{i,g} \geq p_g \\ 0 & \text{if } b_{i,g} < p_g \end{cases} \tag{4}$$

Note that we define $\mathcal{AF}_g(b_{i,g}, x^{([M])})$ to be $x_g^{([M])}$ rather than $x^{([M])}$ when $b_{i,g} \geq p_g$ for ease of presentation due to the zero inter-group transfer assumption. For notational clarity, we define, for any coalition $T \subseteq [M]$,

$$\tilde{x}^{(T)} \triangleq \mathcal{AF}_g(b_{i,g}, x^{(T)}). \tag{5}$$

Having allocated training samples for machine learning to buyer $i$'s prediction task for group $g$, the marketplace then carries out the machine learning to produce predictions and sets a price for the predictions of

$$\mathcal{RF}_{i,g}(b_{i,g}) \triangleq p_g \mathcal{G}(\tilde{x}^{([M])}). \tag{6}$$

Note that this is the revenue function required by Myerson's mechanism for the allocation function $\mathcal{AF}_g$.

Finally, the marketplace divides the collected revenues between the sellers using the Shapley value. For notational clarity, we define $c_T \triangleq |T|!(M - |T| - 1)!/M!$ for any coalition $T \subseteq [M]$. For each buyer $i$ and group $g$, seller $j$ receives the payment division

$$\mathcal{PD}_{i,g,j}(x^{(j)}) = p_g \sum_{T \subseteq [M] \setminus \{j\}} c_T \cdot \left( \mathcal{G}_{i,g}(\tilde{x}^{(T \cup \{j\})}) - \mathcal{G}_{i,g}(\tilde{x}^{(T)}) \right). \tag{7}$$

## 3.4 Market outcomes: utilities and equilibria

We model the data market as a simultaneous game and study its Nash equilibria.

---

*Utilities.* Agents in the data market are strategic and act to maximize their utilities. The marketplace chooses the reserve-price vector $p$ so as to maximize its total revenues,

$$w(p) \triangleq \sum_{i=1}^{N} \sum_{g \in G} \mathcal{RF}_{i,g}(b_{i,g}). \tag{8}$$

Each seller $j$ produces a dataset $x^{(j)}$ to maximize its profits,

$$v_j(x^{(j)}) \triangleq \left( \sum_{i=1}^{N} \sum_{g \in G} \mathcal{PD}_{i,g,j}(x^{(j)}) \right) - \kappa^{(j)T} x^{(j)}. \tag{9}$$

Each buyer $i$ submits bids $b_i$ to maximize its surplus,

$$u_i(b_i) \triangleq \sum_{g \in G} \mu_{i,g} \mathcal{G}_{i,g}(\tilde{x}^{([M])}) - \mathcal{RF}_{i,g}(b_{i,g}) \tag{10}$$

*Nash Equilibrium.* A strategy profile $(p, \{b_i\}, \{x^{(j)}\})$ is a Nash equilibrium if no agent can improve its utility by a unilateral deviation in its strategy.

## 3.5 Fairness in the data market

The fairness model performs three critical functions. First, it provides a criterion that captures the notion of fairness that we study and allows one to test whether fairness has been achieved. Second, it stipulates a particular intervention that the marketplace can undertake to achieve fairness in the intervention scenario. And, third, it allows the cost of the fairness intervention to be assessed. We develop each in turn below.

*Fairness Criterion.* There are many fairness criteria. Which is appropriate depends on the context of an application. Here, we explore just one fairness criterion that is based on the fraction of samples associated to each group in a dataset, i.e., the dataset demographics.

**Definition 3.5.** *(Dataset demographics) The demographics of a dataset $x$ is the vector $\gamma(x) \in \mathbb{R}^{|G|}$ whose $g$-th coordinate, $\gamma_g(x)$ is given by,*

$$\gamma_g(x) \triangleq \frac{x_g}{\|x\|}. \tag{11}$$

Dataset demographics are important for fairness. Sufficiently balanced demographics are important for conducting equity assessments [7, 8]. Unbalanced demographics can unfairly favor machine-learning performance on one group over another [7]. Obtaining more training-data samples for a particular group is an intervention to equalize excess errors [14]. Equalizing excess errors is often applied in contexts where penalizing accuracy on any group is considered unethical, even to achieve fairness, such as in healthcare [14, 40]. Implicit here is a notion that some demographics are balanced, and therefore may be considered fair, while others are unbalanced, and may be considered unfair. We formalize this as follows.

**Definition 3.6.** *(Demographic balance) Let $\gamma \in [0, 1]^{|G|}$ be a target vector satisfying $\sum_{g \in G} \gamma_g = 1$. We say that a dataset $x$ is $\gamma$-demographically balanced if for every $g$ it holds that*

$$\gamma_g = \frac{x_g}{\|x\|}. \tag{12}$$

*Note that we are overloading notation here, we use standalone $\gamma$ to refer to a target vector and function invocation $\gamma(x)$ to refer to the demographics of the dataset $x$.*

*Fairness Intervention.* The marketplace implements the following intervention. The marketplace chooses a target vector $\gamma$. When the sellers submit their datasets, the marketplace accepts a seller $j$'s dataset $x^{(j)}$ if and only if $x^{(j)}$ is $\gamma$-demographically balanced.

If a seller produces any data at equilibrium, its dataset will be demographicaly balanced. Therefore, the aggregate dataset will be demographically balanced. Each seller $j$'s decision is reduced to the total number of samples it will produce, i.e., $\|x^{(j)}\|$. For notational convenience, we define $n^{(j)} \triangleq \|x^{(j)}\|$ for a single seller $j$ and $n^{(T)} \triangleq \sum_{k \in T} n^{(k)}$ for a coalition of sellers $T$. In particular, $n^{([M])} = \|x^{([M])}\|$ is the total number of samples produced by all the sellers in the aggregate dataset.

## 4 DATA MARKET EQUILIBRIA

In this section we study the data market equilibria when the number of buyers is fixed in two scenarios. In the *baseline scenario* the marketplace does not implement its fairness intervention. In the *intervention scenario* the marketplace implements its fairness intervention. We compare the outcomes in the two scenarios to investigate the impacts of the fairness intervention in Sections (5) and (6).

We first show, that solving the equilibria in closed form is elusive in the general case, even under our simplifying assumptions. We therefore focus on the interesting case of *quasi-symmetric setting*.

**Proposition 4.1.** *There does not exist a general closed-form solution over all the possible equilibrium equations in the general setting of the model.*

The proof is in Appendix (A.1). The impossibility arises when prediction tasks can differ in their learnability *within* a group, as captured by the decay parameters $\beta_{i,g}$.

**Definition 4.1.** *(Quasi-symmetric setting) The buyers share a common prediction task within groups and between groups, denoted $\mathcal{G}(\cdot)$, and described by parameters $Z$, $\alpha$, and $\beta$, i.e. for all $i \in [N], g \in G, n \in \mathbb{R}, \mathcal{G}_{i,g}(n) = \mathcal{G}(n)$.*
*The sellers share a common cost structure denoted $\kappa$, i.e., for every pair of sellers $j, j' \in [M], \kappa^{(j)} = \kappa = \kappa^{(j')}$.*

Importantly, although this requires symmetry on the buyers' prediction tasks that capture the learning aspect, we still allow for the values-of-accuracy to vary arbitrarily among buyers and groups to describe a range of economic scenarios (see below). We now describe the buyers' and marketplace's equilibrium strategies before analyzing the sellers' equilibrium stratgies in each scenario.

*Buyer's Dominant Strategy.* Since the marketplace's allocation function and revenue function are an application of Myerson's payment function [2, 38], it follows that truthfulness is a dominant strategy for the buyers.

**Fact 4.1.** *(Buyer Truthfulness) For every buyer $i$, truthfully bidding its values-of-accuracy, i.e., $b_i = \mu_i$ is a dominant strategy.*

Henceforth, we restrict our attention to strategy profiles in which all the buyers bid truthfully, i.e., of the form $\sigma = (p, \{\mu_i\}, \{x^{(j)}\})$.

*Marketplace Best Response.* The marketplace has a best response that depends only on the buyers' strategies.

Let $\sigma = (p, \{\mu_i\}, \{x^{(j)}\})$ be a strategy profile. Straightforward substitution and algebraic manipulation show that the marketplace's utility is given by,

$$w(p) = \sum_{g \in G} \left( p_g \sum_{i=1}^{N} \mathbb{1}[\mu_{i,g} \geq p_g] \right) \mathcal{G}(x_g^{([M])}). \qquad (13)$$

Equation (13) shows that the marketplace extracts revenues from each group independently of the others. The revenues extracted from each group $g$ is the product of two factors: one factor is the accuracy $\mathcal{G}(x^{([M])})$; and the other factor will turn out to be critical in our analyses.

**Definition 4.2.** *(Potential economic value) Let $\sigma = (p, \{\mu_i\}, \{x^{(j)}\})$ be a strategy profile and $g$ be any group in $G$. The* potential economic value *of group $g$ in the strategy profile $\sigma$, denoted $\rho_g$, is defined to be*

$$\rho_g \triangleq p_g \left( \sum_{i=1}^{N} \mathbb{1}[\mu_{i,g} \geq p_g] \right). \qquad (14)$$

The potential economic value, $\rho_g$ is the product of the reserve price and the number of buyers who bid at least the reserve price.

**Fact 4.2.** *(Marketplace Best Response) Let the buyers bid their values-of-accuracy, i.e., $b_i = \mu_i$ for all $i \in [N]$. Let the sellers' datasets $\{x^{(j)}\}, j \in [M]$, be arbitrary. Then, the marketplace's best response is to maximize $\rho_g$ for every group $g \in G$, i.e., to set reserve prices $p_g$ to,*

$$p_g \in \arg\max_{p} \rho_g. \qquad (15)$$

### 4.1 Baseline Scenario Equilibrium

We now analyze the sellers' equilibrium strategies in the baseline scenario, when the marketplace does not constrain their production decisions. We determine the amount of data that the sellers produce, when they produce data.

**Lemma 4.1.** *(Baseline Data Production) If $\sigma = (p, \{\mu_i\}, \{x^{(j)}\})$ is a Nash equilibrium at which samples are produced for some group $g$, i.e., $x_g^{([M])} > 0$, then every seller $j$ produces $x_g^{(j)}$ samples given by,*

$$x_g^{(j)} = \frac{1}{M} \left( \frac{\rho_g}{\kappa_g} \alpha\beta \right)^{1/(\beta+1)}, \qquad (16)$$

*and the total number of samples produced over all the sellers is,*

$$x_g^{([M])} = \left( \frac{\rho_g}{\kappa_g} \alpha\beta \right)^{1/(\beta+1)}. \qquad (17)$$

The proof is in Appendix (A.3). Lemma (4.1) indicates that when the sellers produce data, they produce more data for groups with greater potential economic value, $\rho_g$, and lower production costs, $\kappa_g$. Lemma (4.1) applies when the sellers produce data. But using it, we can characterize the conditions under which the sellers produce data.

**Claim 4.1.** *(Sellers' Baseline Participation Threshold)*
*Let $\sigma = (p, \{\mu_i\}, \{x^{(j)}\})$ be a Nash equilibrium, and $g$ be any group in $G$. The sellers produce a positive number of samples for group $g$,*

i.e., $x_g^{([M])} > 0$, if and only if $\kappa_g \leq \tau_g$, where $\tau_g$ is a threshold value given by

$$\tau_g \triangleq \rho_g c_{\mathcal{G}}, \tag{18}$$

where

$$c_{\mathcal{G}} \triangleq \frac{Z^{\frac{\beta+1}{\beta}}}{\alpha^{\frac{1}{\beta}} \left( \beta^{-\frac{\beta}{\beta+1}} + \beta^{\frac{1}{\beta+1}} \right)^{\frac{\beta+1}{\beta}}}. \tag{19}$$

The proof is in Appendix (A.5). Claim (4.1) indicates that the seller's participation at Nash equilibrium depends critically on the relation between $\kappa_g$ and $\tau_g$. Putting it all together, we can describe the baseline scenario equilibrium as follows.

**Theorem 4.1.** *If $(p, \{\mu_i\}, \{x^{(j)}\})$ is a Nash equilibrium, then for each group $g$, the marketplace sets price $p_g$ to maximize $\rho_g$ and the sellers produce samples depending on the following inequality,*

$$\kappa_g \leq \tau_g. \tag{20}$$

*If Inequality (20) holds, every seller $j$ produces*

$$x_g^{(j)} = \frac{1}{M} \left( \frac{\rho_g}{\kappa_g} \alpha\beta \right)^{1/(\beta+1)} \tag{21}$$

*samples of group $g$, and otherwise $x_g^{(j)} = 0$.*

Theorem 4.1 indicates that samples are produced for the groups independently of each other. Some groups may have zero samples produced. Other groups may have differing numbers of samples produced. All of this depends on each group's economic potential. Distinguishing these possibilities will be important in the next section. Theorem (4.1) also enables us to describe the aggregate-dataset demographics that hold at equilibrium.

**Corollary 4.1.** *Let $(p, \{\mu_i\}, \{x^{(j)}\})$ be a Nash equilibrium, and $H = \{g \in G : \kappa_g \leq \tau_g\}$. Then for every group $g$, if $g \in H$, then,*

$$\frac{x_g^{([M])}}{\|x^{([M])}\|} = \frac{\left( \frac{\rho_g}{\kappa_g} \right)^{1/(\beta+1)}}{\sum_{h \in H} \left( \frac{\rho_h}{\kappa_h} \right)^{1/(\beta+1)}}, \tag{22}$$

*and $\frac{x_g^{([M])}}{\|x^{([M])}\|} = 0$ otherwise.*

Corollary (4.1) reveals the dynamics of data production at baseline equilibrium. Economic disparities drive disparities in the dataset demographics. But the effect is dampened by the diminishing gains in accuracy as the amount of training data grows. These findings illuminate the dynamics of the data market in the baseline scenario that can lead to unfair data production at equilibrium.

## 4.2 Intervention Scenario Equilibrium

We now analyze the sellers' equilibrium strategies in the intervention scenario, when the marketplace constrains the sellers' production decisions. Our goal is to understand how this impacts the levels of data production and the sellers' decisions to participate in the data market. We determine the amount of data that the sellers produce, when they produce data.

**Lemma 4.2.** *(Intervention Data Production) Fix a target vector $\gamma$. If $\sigma = (p, \{\mu_i\}, \{x^{(j)}\})$ is a Nash equilibrium such that samples are produced, i.e. $n^{([M])} > 0$, then there exists a group $h \in G$ such that every seller $j$ produces*

$$n^{(j)} = \frac{1}{M} \left( \frac{\alpha\beta}{\kappa^T\gamma} \sum_{g \in H} \rho_g \gamma_g^{-\beta} \right)^{1/(\beta+1)} \tag{23}$$

*samples, where $H \triangleq \{g \in G : \gamma_g \geq \gamma_h\}$, \hspace{1em} (24)*

*and $\gamma_h$ is a minimum value over $\gamma_g$ satisfying*

$$M\gamma_g n^{(j)} > \left( \frac{\alpha}{Z} \right)^{\frac{1}{\beta}}. \tag{25}$$

The proof is in Appendix (A.7). Lemma (4.2) shows how coupling data production across the groups via $\gamma$ affects data production. The sellers produce more data as the groups' potential economic values increase, but this is mediated by their required representation, $\rho_g \gamma_g^{-\beta}$. And the sellers produce more data as the marginal production cost $\kappa^T\gamma$ decreases. We can use Lemma (4.2) to give a necessary condition for the sellers to produce data.

**Claim 4.2.** *(Sellers' Intervention Participation Thresholds) Fix a target vector $\gamma$. Let $\sigma = (p, \{\mu_i\}, \{x^{(j)}\})$ be a Nash equilibrium. If the sellers produce a positive number of samples, i.e., $n^{([M])} > 0$, then there exists a group $h \in G$ such that the sellers' marginal production cost, $\kappa^T\gamma$, is at most a threshold value, $\tau_H(\rho, \gamma)$, given by,*

$$\tau_H(\rho, \gamma) = \frac{\left( \sum_{g \in H} \rho_g \right)^{\frac{\beta+1}{\beta}}}{\left( \sum_{g \in H} \rho_g \gamma_g^{-\beta} \right)^{\frac{1}{\beta}}} \cdot c_{\mathcal{G}} \tag{26}$$

*where $H \triangleq \{g \in G : \gamma_g \geq \gamma_h\}$, \hspace{1em} (27)*

*and $\gamma_h$ is a minimum value over $\gamma_g$ satisfying*

$$\gamma_g n^{([M])} > \left( \frac{\alpha}{Z} \right)^{\frac{1}{\beta}}. \tag{28}$$

The proof is in Appendix (A.8). Claim (4.2) tells us that if the sellers produce data at Nash equilibrium in the intervention scenario, then they must be profitably monetizing some groups. Altogether, we have the following description of the intervention scenario equilibrium.

**Theorem 4.2.** *Let $(p, \{\mu_i\}, \{x^{(j)}\})$ be a Nash equilibrium, then for each group $g$, the marketplace sets reserve price $p_g$ to maximize $\rho_g$ and each seller $j$ produces data if and only if the sellers' marginal production cost, $\kappa^T\gamma$, is at most a threshold value, $\tau_H(\rho, \gamma)$, given by,*

$$\tau_H(\rho, \gamma) = \frac{\left( \sum_{g \in H} \rho_g \right)^{\frac{\beta+1}{\beta}}}{\left( \sum_{g \in H} \rho_g \gamma_g^{-\beta} \right)^{\frac{1}{\beta}}} \cdot c_{\mathcal{G}} \tag{29}$$

*samples, where $H \triangleq \{g \in G : \gamma_g \geq \gamma_h\}$, \hspace{1em} (30)*

*for some group $h$ such that $\gamma_h$ is a minimum value over $\gamma_g$ satisfying*

$$\gamma_g n^{([M])} > \left( \frac{\alpha}{Z} \right)^{\frac{1}{\beta}}. \tag{31}$$

# 5 INTERVENTION CAN PREVENT FORMATION IN EMERGING MARKETS

The emergence of ethical profit-seeking firms in online data markets raises a fundamental question: Why now? Online data markets have existed since at least 2009 [47]. Interest in ethical issues related to data date back to at least the 1970's [17]. Why would these firms emerge in this moment, and not earlier or later?

In this section we get at this question by considering emerging markets, i.e., markets where the economic potential is small or not yet fully realized, and data production has not yet begun or is still low. What happens if a fairness constraint is imposed from the get-go? Can the market overcome the cost of fairness and still incentivize sellers to participate? Or will the cost of fairness drive the sellers out of the market?

Formally, we study the conditions under which the sellers produce data in the baseline scenario but not in the intervention scenario. To make this precise we define market formation and intervention backfire.

**Definition 5.1.** *(Market Formation) Fix a set of $N$ buyers and $M$ sellers. In either the baseline or intervention scenario, we say the market forms if there exists a Nash equilibrium, $\sigma = (p, \{\mu_i\}, \{x^{(j)}\})$, such that samples are produced, i.e., $\|x^{([M])}\| > 0$.*

**Definition 5.2.** *(Intervention backfire) Let $\gamma$ be a target vector, and fix a data market. We say that the intervention $\gamma$ backfires in the data market if the market forms in the baseline scenario but not in the intervention scenario.*

**Theorem 5.1.** *For every target vector $\gamma$ there exists a data market in which $\gamma$ backfires.*

The proof is in Appendix (A.9), and follows from constructing a market in which the production cost for one group is arbitrarily high relative to its economic potential. Theorem (5.1) tells us that all interventions are risky when the intervention requires the sellers to begin producing data for some group. If the marketplace does not choose the target vector $\gamma$ carefully, then the sellers may opt out of the market.

This is striking because this magnifies the cost of fairness *to the maximum extent possible and to every single agent*, i.e., the cost of fairness for every agent is its full baseline utility. And this can harm the very groups that are the intended beneficiaries of the intervention. Some groups may be better off with some samples in the baseline scenario—even if they are under-represented—versus no samples in the intervention scenario.

Yet, Theorem (5.1) is tempered by its assumption that no data is produced for some of the groups in the baseline scenario. It is sensible to ask whether the backfire risk behaves differently when data is produced for all the groups in the baseline scenario.

**Definition 5.3.** *(Fully-forming Markets) Fix a set of $N$ buyers and $M$ sellers. We say the market fully-forms if there exists a Nash equilibrium in the baseline scenario, $\sigma = (p, \{\mu_i\}, \{x^{(j)}\})$, such that samples are produced for every group, i.e., for all $g \in G$, $\|x_g^{([M])}\| > 0$.*

In contrast to the general case, our next result shows there exists a target vector that never backfires in fully-forming markets.

**Definition 5.4.** *(Uniform intervention) The uniform intervention, denoted $u$, is the target vector $u_g \triangleq 1/|G|$, for every group $g$.*

**Theorem 5.2.** *Let $N$ buyers and $M$ sellers be a fully-forming data market. If the marketplace chooses the uniform intervention, i.e., $\gamma = u$, then the data market forms in the intervention scenario.*

The proof is in Appendix (A.10). It is suprising that there exists a target vector that never backfires in fully-forming markets. It is natural to ask: How much flexibility does the marketplace have in fully-forming markets?

**Theorem 5.3.** *Let $\gamma$ be the marketplace's target vector. If $\gamma$ is not the uniform intervention, i.e., $\gamma \neq u$, then there exists a data-market that is fully forming in the baseline scenario but does not form in the intervention scenario.*

The proof is in Appendix (A.11). Theorem (5.3) clarifies that the backfire risk is still present in fully-forming markets. Theorems (5.2) and (5.3) underscore the question: How can the backfire risk be mitigated in fully-forming markets? We next give sufficient conditions for a fully-forming market that ensure the market will form in the intervention scenario.

**Theorem 5.4.** *Let $N$ buyers and $M$ sellers be a fully-forming data market. Define $\eta \in [0, 1]$ to be the minimum value satisfying for all $g \in G$,*

$$\kappa_g \leq \eta \tau_g. \tag{32}$$

*Let $\gamma$ be an intervention. Define $a \geq 1$,*

$$\frac{1}{a} = \min_{g \in G} \gamma_g, \tag{33}$$

*and $b \geq 1$*

$$\frac{1}{b} = \max_{g \in G} \gamma_g. \tag{34}$$

*If the marketplace chooses target vector $\gamma$ and*

$$\eta < \left(\frac{b}{a}\right)^{\beta+1} \frac{1}{r|G|}, \tag{35}$$

*where $r$ is a constant that depends on the $N$ buyers, then the market will form in the intervention scenario.*

The proof is in Appendix (A.12). Altogether, our results on the backfire risk suggest that, in emerging markets, ethical firms may not be economically viable or may lack sufficient flexibility to act on their ethical objectives. This suggests that ethical firms may not have emerged earlier because the economic potential of data markets was insufficient to support them.

# 6 MARKET GROWTH CAN AMORTIZE THE COST OF FAIRNESS IN ESTABLISHED MARKETS

Our results on the backfire risk suggest an explanation for why ethical firms have not emerged sooner. But what has changed that is conducive to the emergence of ethical firms now? In this section, we study one possibility: data markets may now be sufficiently large and established. Formally, we study market growth.

We model market growth as markets with more buyers. For a fixed number of sellers $M$, a sequence of values-of-accuracy vectors

$(\mu_N)_{N \in \mathbb{N}}$ defines a sequence of buyers and in turn a sequence of data markets. The $N$-th data market is defined by the $M$ sellers and the first $N$ buyers in the buyers sequence. We examine the Nash equilibrium outcomes of the individual data markets as well as in the limit as an unbounded number of buyers enter the data market.

Market growth turns out to be sufficient to mitigate the backfire risk.

**Claim 6.1.** *Let $\gamma$ be the marketplace's target vector. If there exists $g \in G$ such that $\max_{p_g} \rho_g \to \infty$ as $N \to \infty$, then there exists an $N_0$ such that $N > N_0$ implies that for all $j$, $\|y^{(j)}\| > 0$.*

The proof is in Appendix (A.13). Claim (6.1) indicates that the ability of market growth to mitigate the backfire risk is flexible. It only requires the economic potential of *a single group* to be sufficiently large. The intervention can then redirect value extracted from one group towards data production for another group.

After the backfire risk is successfully mitigated, there will still be a cost of fairness, therefore it is natural to ask: What happens to the cost of fairness as the potential economic value of the market continues to grow?

The cost of fairness quantifies the burden imposed by a fairness intervention. It is typically defined as the difference between an agent's utility without fairness requirements and its utility with fairness requirements. This definition is not suitable for our analysis because: 1) we find that some agents can be strictly better off in the intervention scenario; and 2) the cost of fairness is in absolute terms which may be misleading in comparing markets of different size.

Therefore we focus on the ratio of an agent's intervention utility to its baseline utility. This ratio is directly related to the cost of fairness, and it captures the burden of a fairness intervention in normalized terms.

Let $(p, \{b_i\}, \{x^{(j)}\})$ be a Nash equilibrium in the baseline scenario, and $(p^f, \{b_i^f\}, \{y^{(j)}\})$ be a Nash equilibrium in the intervention scenario. The marketplace's utility ratio is,

$$UR_{Mkt}(p, p^f) \triangleq \frac{w^f(p^f)}{w(p)}, \tag{36}$$

where $w^f(p^f)$ is the marketplace's utility in the intervention scenario. Seller $j$'s utility ratio is,

$$UR_{S,j}(x^{(j)}, y^{(j)}) \triangleq \frac{v_j^f(y^{(j)})}{v_j(x^{(j)})}, \tag{37}$$

where $v_j^f(y^{(j)})$ is seller $j$'s utility in the intervention scenario. Buyer $i$'s utility ratio is,

$$UR_{B,i}(b_i, b_i^f) \triangleq \frac{u_i^f(b_i^f)}{u_i(b_i)}, \tag{38}$$

where $u_i^f(b_i^f)$ is buyer $i$'s utility in the intervention scenario.

Market growth also suffices to attenuate the cost of fairness in normalized terms.

**Theorem 6.1.** *If there exists $g \in G$ such that $\max_{p_g} \rho_g \to \infty$ as $N \to \infty$, then for the marketplace we have*

$$\lim_{N \to \infty} \frac{w^f(p)}{w(p)} = 1, \tag{39}$$

*for every seller $j$ we have*

$$\lim_{N \to \infty} \frac{v_j^f(y^{(j)})}{v_j(x^{(j)})} = 1, \tag{40}$$

*and for every buyer $i$ we have*

$$\lim_{N \to \infty} \frac{u_i^f(\mu_i)}{u_i(\mu_i)} \geq 1. \tag{41}$$

The proof is in Appendix (A.17). Perhaps surprisingly, if the potential economic value of at least one group grows unbounded as buyers enter the market, then *every agent in the data market is asymptotically at least as well off* in the intervention scenario than in the baseline scenario, and sometimes some of the buyers can be strictly better off. Stated another way: market growth can amortize the cost of fairness, for any given group fairness either has a vanishing cost or creates a positive externality.

Altogether, our results on market growth suggest that market conditions may have become more hospitable to ethical firms. Surging demand for data may have driven market growth to a level that can economically sustain them.

## 7 LIMITATIONS AND DISCUSSION

Our results come with some important limitations.

- The fairness intervention that we study is naive. For instance, it does not allow sellers to specialize in producing samples for specific groups while collectively remaining balanced. Our work to reduce cost of fairness even further while addressing it.
- Another limitation is that we study only one fairness concern in this work, that stems from unavailability of representative data. We expect that economic growth can dampens the cost of fairness for other criteria.
- Assuming no transfer of knowledge for prediction between groups is an extreme case. It is interesting that it does not prevent fairness to trickle down with market growth. Still we expect intermediate regimes of transfer in practice [3, 25] and would like to leverage it for emerging or mid-size data markets.
- We focus here on one comprehensive and often cited data-market model [2]. Although it captures multiple aspects of significant real-world and theoretical data markets, our results will vary when qualitatively different incentive structure are offered to sellers and buyers.

Notwithstanding these limitations, we believe that our work points to exciting opportunities. Data markets are often praised for their superior ability to extract value from data but criticized for the ethical concerns they raise including fairness. So far fairness has rarely been considered except as an obstacle [21, 34]. Our results suggest that value extraction and fairness are not always at odds: at least one established model of data markets aligns efficient value extraction with a higher mandate to ensure fairness conditions and convert economic growth into opportunities to intervene for fairness. We believe that our results may be only the first in that direction; other market mechanisms and other forms of fairness or ethical objectives could also be leveraging interventions that take into account the market's endogenous response.

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

## A PROOFS

### A.1 Proof of Proposition (4.1)

**Proposition 4.1.** *There does not exist a general closed-form solution over all the possible equilibrium equations in the general setting of the model.*

Proof. Consider the seller's marginal utility at equilibrium.

**Observation A.1.** *Fix the bid $b_i$ of each buyer $i \in [N]$, the marketplace's posted price vector $p$, and the dataset $x^{(k)}$ of each seller $k \in [M] \setminus \{j\}$. Let $x^{(j)}$ be seller $j$'s best response in the baseline scenario. For each group $g$, if $x_g^{(j)} > 0$, then $x_g^{(j)}$ satisfies the equation,*

$$p_g \sum_{i \in B_g} \sum_{T \subseteq [M] \setminus \{j\}} c_T \alpha_{i,g} \beta_{i,g} (x_g^{(T)} + x_g^{(j)})^{-\beta_{i,g}-1} = \kappa_g^{(j)}, \quad (42)$$

*where $B_g \triangleq \{i \in [N] : b_{i,g} \geq p_g\}$ is the set of buyers that bid at least $p_g$ for group $g$.*

Equation (42) indicates that the set of possible equilibrium equations includes a large class of polynomial equations that includes polynomial equations of arbitrarily large degree.

The celebrated Abel–Ruffini Theorem [1] tells us that there is no general closed-form solution to polynomial equations of degree five or higher. We cannot immediately apply the theorem, however, because the coefficients that may appear in the equilibrium polynomial equations are not unrestricted. Do polynomial equations without closed-form solutions occur in the equilibrium equations in the general setting of the model? Yes, as demonstrated by the following toy example.

Example A.1. *(van der Waerden's Quintic) Consider the following instance of the model. Let there be 2 buyers and 1 seller. For some group $g$, set the parameters as follows: $\mu_{1,g} = 1 = \mu_{2,g}$, $Z_{1,g} = 5 = Z_{2,g}$, $\alpha_{1,g} = 1/3$, $\beta_{1,g} = 3$, $\alpha_{2,g} = 1/4$, $\beta_{2,g} = 4$, and $\kappa_g^{(1)} = 1$.*

*We know that in the baseline, the marketplace and seller will choose their strategies independently for each group. Since there are 2 buyers each with the same value-of-accuracy for this group, at equilibrium, the marketplace will set $p_g = \mu_{1,g} = \mu_{2,g} = 1$. And since there is only one seller, Equation (42) becomes*

$$\sum_{i \in \{1,2\}} \alpha_{i,g} \beta_{i,g} (x_g^{(1)})^{-\beta_{i,g}-1} = \kappa_g^{(1)}. \quad (43)$$

*Multiplying both sides by $(x_g^{(1)})^5$ and rearranging yields the following polynomial equation:*

$$(x_g^{(1)})^5 - (x_g^{(1)}) - 1 = 0, \quad (44)$$

*which is an example given in [51] as having no closed-form solution. Still, the seller will only solve Equation (44) exactly at equilibrium if it can achieve positive utility. The root of Equation (44) is approximately 1.1673, thus we can bound the utility the seller will receive from group $g$ by the prediction gain of producing 1 sample and the cost of*

*producing 2 samples*

$$v_1(x^{(1)}) = \sum_{i \in \{1,2\}} \sum_{h \in G} p_h \mathcal{G}(x_h^{(1)}) - \kappa_h^{(1)} x_h^{(1)} \quad (45)$$

$$\geq \sum_{i \in \{1,2\}} \mathcal{G}_{i,g}(1) - 2 \quad (46)$$

$$= \left(5 - \frac{1}{3}\right) + \left(5 - \frac{1}{4}\right) - 2 > 0. \quad (47)$$

*We conclude that at equilibrium $x_g^{(1)} > 0$, hence the seller solves Equation (44).*

□

### A.2 Proof of Fact (A.1)

**Fact A.1.** *If $\sigma = (p, \{\mu_i\}, \{x^{(j)}\})$ is a Nash equilibrium, then for every $j, j' \in [M]$ we have that $x^{(j)} = x^{(j')}$.*

Proof. Seller $j$'s utility is given by

$$v_j(x^{(j)}) \triangleq \left( \sum_{i=1}^{N} \sum_{g \in G} \mathcal{P}\mathcal{D}_{i,g,j}(x^{(j)}) \right) - \kappa^{(j)T} x^{(j)} \quad (48)$$

$$= \sum_{g \in G} \rho_g \sum_{T \subseteq [M] \setminus \{j\}} c_T \left( \mathcal{G}(x_g^{(T \cup \{j\})}) - \mathcal{G}(x_g^{(T)}) \right) \quad (49)$$

$$- \sum_{g \in G} \kappa_g x_g^{(j)}. \quad (50)$$

where $\mathbf{1}[\cdot]$ is the indicator function. At equilibrium, seller $j$'s marginal utility in $x_g^{(j)}$ must be 0 for every group. Moreover, this must also be true for any other seller $k$. Hence, at equilibrium, seller $j$'s marginal utility with respect to $x_g^{(j)}$ must equal seller $k$'s marginal utility with respect to $x_g^{(k)}$ for any two sellers $j$ and $k$.

We now show that if two sellers, $j$ and $k$ produce different amounts of data for any group $g$, $x_g^{(j)} \neq x_g^{(k)}$, then $(p, \{\mu_i\}, \{x^j\})$ is not a Nash equilibrium. By definition, seller $j$'s marginal utility with respect to $x_g^{(j)}$ is

$$\frac{\partial}{\partial x_g^{(j)}} v_j(x^{(j)}) = \rho_g \left( \sum_{T \subseteq [M] \setminus \{j\}} c_T \mathcal{G}'(x_g^{(T \cup \{j\})}) \right) - \kappa_g. \quad (51)$$

We want to write seller $j$'s marginal utility in a form that can be easily compared with seller $k$'s marginal utility. We can accomplish this by changing the index set of the summation from $[M] \setminus \{j\}$ to $[M] \setminus \{j, k\}$ as follows,

$$\frac{\partial}{\partial x_g^{(j)}} v_j(x^{(j)}) = \rho_g \sum_{T \subseteq [M] \setminus \{j,k\}} c_T \mathcal{G}'(x_g^{(T \cup \{j\})})$$
$$+ c_{T \cup \{k\}} \mathcal{G}'(x_g^{((T \cup \{k\}) \cup \{j\})}) - \kappa_g.$$

Similarly, write seller $k$'s marginal utility with respect to $x_g^{(k)}$ as

$$\frac{\partial}{\partial x_g^{(k)}} v_k(x^{(k)}) = \rho_g \sum_{T \subseteq [M] \setminus \{j,k\}} c_T \mathcal{G}'(x_g^{(T \cup \{k\})})$$
$$+ c_{T \cup \{k\}} \mathcal{G}'(x_g^{((T \cup \{j\}) \cup \{k\})}) - \kappa_g.$$

Note that $\mathcal{G}'$ is strictly decreasing. Without loss of generality, assume that $x_g^{(j)} > x_g^{(k)}$, then for every $T \subseteq [M] \setminus \{j, k\}$ it holds that

$$\mathcal{G}'(x_g^{(T \cup \{j\})}) < \mathcal{G}'(x_g^{(T\{k\})}),$$

and

$$\mathcal{G}'(x_g^{((T \cup \{k\}) \cup \{j\})}) = \mathcal{G}'(x_g^{((T \cup \{j\}) \cup \{k\})}).$$

Consequently, the derivative of seller $j$'s utility is strictly less than that of seller $k$'s. We conclude $\sigma$ is not a Nash equilibrium. $\square$

## A.3 Proof of Lemma (4.1)

**Lemma 4.1.** *(Baseline Data Production) If $\sigma = (p, \{\mu_i\}, \{x^{(j)}\})$ is a Nash equilibrium at which samples are produced for some group $g$, i.e., $x_g^{([M])} > 0$, then every seller $j$ produces $x_g^{(j)}$ samples given by,*

$$x_g^{(j)} = \frac{1}{M}\left(\frac{\rho_g}{\kappa_g}\alpha\beta\right)^{1/(\beta+1)}, \tag{16}$$

*and the total number of samples produced over all the sellers is,*

$$x_g^{([M])} = \left(\frac{\rho_g}{\kappa_g}\alpha\beta\right)^{1/(\beta+1)}. \tag{17}$$

PROOF. Seller $j$'s utility is given by

$$v_j(x^{(j)}) = \left(\sum_{i=1}^{N}\sum_{g \in G}\mathcal{PD}_{i,g,j}(x^{(j)})\right) - \kappa^{(j)T}x^{(j)} \tag{52}$$

$$= \sum_{g \in G}\rho_g\sum_{T \subseteq [M]\setminus\{j\}}c_T\left(\mathcal{G}(x_g^{(T \cup \{j\})}) - \mathcal{G}(x_g^{(T)})\right)$$

$$- \sum_{g \in G}\kappa_g x_g^{(j)}. \tag{53}$$

By Fact (A.1) the sellers all make the same production decisions at equilibrium, i.e. for every pair of sellers $j, k \in [M]$ we have $x^{(j)} = x^{(k)}$. Consequently, every seller makes the same average marginal contribution over all the coalitions, so the sellers split the revenues collected from each buyer evenly. Therefore, seller $j$'s utility is

$$v_j(x^{(j)}) = \frac{1}{M}\sum_{g \in G}\rho_g\mathcal{G}(x_g^{([M])}) - \kappa_g x_g^{(j)}. \tag{54}$$

Fact (A.1) also implies that $x_g^{([M])} = Mx_g^{(j)}$, thus

$$v_j(x^{(j)}) = \frac{1}{M}\sum_{g \in G}\rho_g\mathcal{G}(Mx_g^{(j)}) - \kappa_g x_g^{(j)}. \tag{55}$$

Seller $j$'s marginal utility in $x_g^{(j)}$ is therefore,

$$\frac{\partial}{\partial x_g^{(j)}}v_j(x^{(j)}) = \frac{1}{M}\rho_g\left(\frac{\partial}{\partial x_g^{(j)}}\mathcal{G}(Mx_g^{(j)})\right) - \kappa_g. \tag{56}$$

Now,

$$\frac{\partial}{\partial x_g^{(j)}}\mathcal{G}(Mx_g^{(j)}) = \begin{cases} 0 & \text{if } Mx_g^{(j)} < (\alpha/Z)^{1/\beta} \\ \alpha\beta M^{-\beta}(x_g^{(j)})^{-\beta-1} & \text{if } (\alpha/Z)^{1/\beta} < Mx_g^{(j)} \end{cases} \tag{57}$$

If $Mx_g^{(j)} < (\alpha/Z)^{1/\beta}$, then seller $j$'s marginal utility is negative. By assumption, $\sigma$ is a Nash equilibrium at which the sellers produce

data. At equilibrium, seller $j$'s marginal utility in each $x_g^{(j)}$ must be 0. So we must have $Mx_g^{(j)} > (\alpha/Z)^{1/\beta}$, and seller $j$'s marginal utility is

$$\frac{\partial}{\partial x_g^{(j)}}v_j(x^{(j)}) = \frac{1}{M}\rho_g\alpha\beta M^{-\beta}(x_g^{(j)})^{-\beta-1} - \kappa_g. \tag{58}$$

Setting this equal to 0 and solving for $x_g^{(j)}$ completes the proof. $\square$

## A.4 Proof of Fact (A.2)

**Fact A.2.** *If $\beta > 0$, then*

$$\beta\left(\beta^{-\frac{\beta}{\beta+1}} + \beta^{\frac{1}{\beta+1}}\right)^{\frac{\beta+1}{\beta}} > 1. \tag{59}$$

PROOF. Towards contradiction, suppose that

$$\beta\left(\beta^{-\frac{\beta}{\beta+1}} + \beta^{\frac{1}{\beta+1}}\right)^{\frac{\beta+1}{\beta}} \leq 1. \tag{60}$$

And write,

$$\beta\left(\beta^{-\frac{\beta}{\beta+1}} + \beta^{\frac{1}{\beta+1}}\right)^{\frac{\beta+1}{\beta}} \leq 1 \tag{61}$$

$$\implies \beta^{\frac{\beta}{\beta+1}}\left(\beta^{-\frac{\beta}{\beta+1}} + \beta^{\frac{1}{\beta+1}}\right) \leq 1 \tag{62}$$

$$\implies 1 + \beta \leq 1 \tag{63}$$

$$\implies \beta \leq 0. \tag{64}$$

But $\beta > 0$ by definition of the model, a contradiction. $\square$

## A.5 Proof of Claim (4.1)

**Claim 4.1.** *(Sellers' Baseline Participation Threshold)*
*Let $\sigma = (p, \{\mu_i\}, \{x^{(j)}\})$ be a Nash equilibrium, and $g$ be any group in $G$. The sellers produce a positive number of samples for group $g$, i.e., $x_g^{([M])} > 0$, if and only if $\kappa_g \leq \tau_g$, where $\tau_g$ is a threshold value given by*

$$\tau_g \triangleq \rho_g c_G, \tag{18}$$

*where*

$$c_G \triangleq \frac{Z^{\frac{\beta+1}{\beta}}}{\alpha^{\frac{1}{\beta}}\left(\beta^{-\frac{\beta}{\beta+1}} + \beta^{\frac{1}{\beta+1}}\right)^{\frac{\beta+1}{\beta}}}. \tag{19}$$

PROOF. We first prove the forward direction, i.e., if $x_g^{([M])} > 0$, then $\kappa_g \leq \tau_g$. Fix a seller $j$. By assumption, $\sigma$ is a Nash equilibrium, so every seller produces the same dataset at equilibrium by Fact (A.1). Hence seller $j$'s utility is

$$v_j(x^{(j)}) = \frac{1}{M}\sum_{g \in G}\rho_g\mathcal{G}(Mx_g^{(j)}) - \kappa_g x_g^{(j)}. \tag{65}$$

Also by assumption, $x_g^{([M])} > 0$. Since value is extracted independently by group in the baseline scenario, it must hold that

$$\frac{1}{M}\rho_g\mathcal{G}(Mx_g^{(j)}) - \kappa_g x_g^{(j)} \geq 0, \tag{66}$$

because otherwise seller $j$ could improve its utility by producing no data for group $g$ and $\sigma$ would not be a Nash equilibrium.

 

By Lemma (4.1),

$$x_g^{(j)} = \frac{1}{M}\left(\frac{\rho_g}{\kappa_g}\alpha\beta\right)^{\frac{1}{\beta+1}}. \tag{67}$$

Therefore, we can write Inequality (66) as

$$\rho_g\mathcal{G}\left(\left(\frac{\rho_g}{\kappa_g}\alpha\beta\right)^{\frac{1}{\beta+1}}\right) - \kappa_g\left(\frac{\rho_g}{\kappa_g}\alpha\beta\right)^{\frac{1}{\beta+1}} \geq 0. \tag{68}$$

Recall that $\mathcal{G}$ is defined piecewise, we must ascertain which piece applies, i.e., whether or not,

$$\left(\frac{\rho_g}{\kappa_g}\alpha\beta\right)^{\frac{1}{\beta+1}} \geq \left(\frac{\alpha}{Z}\right)^{\frac{1}{\beta}}. \tag{69}$$

Observe that if Inequality (69) does not hold, then seller $j$ can improve its utility by producing no data for group $g$ and $\sigma$ would not be a Nash equilibrium. Therefore, Inequality (69) holds, and we have

$$\mathcal{G}\left(\left(\frac{\rho_g}{\kappa_g}\alpha\beta\right)^{\frac{1}{\beta+1}}\right) = Z - \alpha\left(\left(\frac{\rho_g}{\kappa_g}\alpha\beta\right)^{\frac{1}{\beta+1}}\right)^{-\beta}. \tag{70}$$

Substituting Equation (70) into Inequality (68) and some straightforward algebra yields:

$$\kappa_g \leq \frac{\rho_g Z^{\frac{\beta+1}{\beta}}}{\alpha^{\frac{1}{\beta}}\left(\beta^{-\frac{\beta}{\beta+1}} + \beta^{\frac{1}{\beta+1}}\right)^{\frac{\beta+1}{\beta}}} = \tau_g. \tag{71}$$

This proves the forward direction since $j$ is arbitrary.

We now prove the reverse direction, i.e., if $\kappa_g \leq \tau_g$, then $x_g^{([M])} > 0$. We must show that the sellers will obtain non-negative utility by producing a positive number of samples, i.e., Inequality (66) holds. To do so, we must evaluate

$$\mathcal{G}\left(\left(\frac{\rho_g}{\kappa_g}\alpha\beta\right)^{\frac{1}{\beta+1}}\right) \tag{72}$$

which depends on whether Inequality (69) holds. We first show that it does.

By assumption, $\kappa_g \leq \tau_g$ and therefore

$$\left(\frac{\rho_g}{\kappa_g}\alpha\beta\right)^{\frac{1}{\beta+1}} \geq \left(\frac{\rho_g}{\tau_g}\alpha\beta\right)^{\frac{1}{\beta+1}} \tag{73}$$

$$= \left(\frac{\alpha}{Z}\right)^{\frac{1}{\beta}}\left(\beta\left(\beta^{-\frac{\beta}{\beta+1}} + \beta^{\frac{1}{\beta+1}}\right)^{\frac{\beta+1}{\beta}}\right)^{\frac{1}{\beta+1}} \tag{74}$$

$$> \left(\frac{\alpha}{Z}\right)^{\frac{1}{\beta}}, \tag{75}$$

since

$$\beta\left(\beta^{-\frac{\beta}{\beta+1}} + \beta^{\frac{1}{\beta+1}}\right)^{\frac{\beta+1}{\beta}} > 1, \tag{76}$$

by Fact (A.2).

Thus, we can write Inequality (66) as

$$\frac{1}{M}\rho_g\left(Z - \alpha\left(\left(\frac{\rho_g}{\kappa_g}\alpha\beta\right)^{\frac{1}{\beta+1}}\right)^{-\beta}\right) - \kappa_g\frac{1}{M}\left(\frac{\rho_g}{\kappa_g}\alpha\beta\right)^{\frac{1}{\beta+1}} \geq 0. \tag{77}$$

Applying the assumption $\kappa_g \leq \tau_g$ and some straightforward algebra complete the proof. $\square$

## A.6 Proof of Fact (A.3)

**Fact A.3.** *If $(p, \{\mu_i\}, \{x^{(j)}\})$ is a Nash equilibrium, then for every $j, k \in [M]$ we have that $x^{(j)} = x^{(k)}$.*

PROOF. In the intervention scenario, the marketplace's intervention couples data production across the groups. Therefore, the production decision that each seller $j$ faces is how many samples in total to produce, i.e., $n^{(j)} \triangleq \|x^{(j)}\|$, because the number of samples of each group, $x_g^{(j)}$, is then determined by the marketplace's choice of target vector $\gamma$, i.e. $x_g^{(j)} = \gamma_g n^{(j)}$. Define $n^{(T)} \triangleq \sum_{k \in T} n^{(k)}$ for any $T \subseteq [M]$ and write seller $j$'s utility by

$$v_j(x^{(j)}) = \sum_{g \in G}\rho_g\sum_{T \subseteq [M]\backslash\{j\}} c_T\left(\mathcal{G}(\gamma_g n^{(T \cup \{j\})}) - \mathcal{G}(\gamma_g n^{(T)})\right)$$
$$- \kappa^T\gamma n^{(j)} \tag{78}$$
$$\triangleq v_j(n^{(j)}). \tag{79}$$

At equilibrium, every seller $j$'s marginal utility must be 0. Therefore, for any two sellers $j$ and $k$, seller $j$'s marginal utility must equal seller $k$'s marginal utility; formally, we must have,

$$\frac{\partial}{\partial n^{(j)}}v_j(n^{(j)}) = 0 = \frac{\partial}{\partial n^{(k)}}v_k(n^{(k)}). \tag{80}$$

We now show that if two sellers, $j$ and $k$ produce a different total number of samples, i.e., $n^{(j)} \neq n^{(k)}$, then $(p, \{\mu_i\}, \{x^{(j)}\})$ is not a Nash equilibrium. Write seller $j$'s marginal utility with respect to $n^{(j)}$ as

$$\frac{\partial}{\partial n^{(j)}}v_j(n^{(j)}) = \sum_{g \in G}\rho_g\sum_{T \subseteq [M]\backslash\{j,k\}} c_T\mathcal{G}'(\gamma_g n^{(T \cup \{j\})})$$
$$+ c_{T \cup \{k\}}\mathcal{G}'(\gamma_g n^{((T \cup \{k\}) \cup \{j\})}) - \kappa^T\gamma. \tag{81}$$

Write seller $k$'s marginal utility with respect to $n^{(k)}$ as

$$\frac{\partial}{\partial n^{(k)}}v_k(n^{(k)}) = \sum_{g \in G}\rho_g\sum_{T \subseteq [M]\backslash\{j,k\}} c_T\mathcal{G}'(\gamma_g n^{(T \cup \{k\})})$$
$$+ c_{T \cup \{k\}}\mathcal{G}'(\gamma_g n^{((T \cup \{j\}) \cup \{k\})}) - \kappa^T\gamma. \tag{82}$$

Note that $\mathcal{G}'$ is strictly decreasing. Without loss of generality and towards contradiction, assume that $n^{(j)} > n^{(k)}$, then for every $T \subseteq [M]\backslash\{j,k\}$ it holds that

$$\mathcal{G}'(\gamma_g n^{(T \cup \{j\})}) < \mathcal{G}'(\gamma_g n^{(T \cup \{k\})}), \tag{83}$$

and

$$\mathcal{G}'(\gamma_g n^{((T \cup \{k\}) \cup \{j\})}) = \mathcal{G}'(\gamma_g n^{((T \cup \{j\}) \cup \{k\})}). \tag{84}$$

Consequently, the derivative of seller $j$'s utility is strictly less than that of seller $k$'s, a contradiction. We conclude that $n^{(j)} = n^{(k)}$, i.e., $x^{(j)} = x^{(k)}$. $\square$

## A.7 Proof of Lemma (4.2)

**Lemma 4.2.** *(Intervention Data Production) Fix a target vector $\gamma$. If $\sigma = (p, \{\mu_i\}, \{x^{(j)}\})$ is a Nash equilibrium such that samples are*

produced, i.e. $n^{([M])} > 0$, then there exists a group $h \in G$ such that every seller $j$ produces

$$n^{(j)} = \frac{1}{M} \left( \frac{\alpha\beta}{\kappa^T \gamma} \sum_{g \in H} \rho_g \gamma_g^{-\beta} \right)^{1/(\beta+1)} \tag{23}$$

samples, where $H \triangleq \{g \in G : \gamma_g \geq \gamma_h\}$, $\tag{24}$

and $\gamma_h$ is a minimum value over $\gamma_g$ satisfying

$$M\gamma_g n^{(j)} > \left( \frac{\alpha}{Z} \right)^{\frac{1}{\beta}}. \tag{25}$$

PROOF. Write seller $j$'s utility as follows

$$v_j(x^{(j)}) = \left( \sum_{i=1}^N \sum_{g \in G} \mathcal{PD}_{i,g,j}(x^{(j)}) \right) - \kappa^{(j)T} x^{(j)} \tag{85}$$

$$= \sum_{i=1}^N \sum_{g \in G} p_g \sum_{T \subseteq [M] \setminus \{j\}} c_T \cdot \mathcal{G}_{i,g}(\mathcal{AF}_g(\mu_{i,g}, x^{(T \cup \{j\})})) $$
$$- \mathcal{G}_{i,g}(\mathcal{AF}_g(\mu_{i,g}, x^{(T)})) - \kappa^{(j)T} x^{(j)} \tag{86}$$

$$= \left( \sum_{i=1}^N \sum_{g \in G} p_g \frac{1}{M} \mathcal{G}_{i,g}(\mathcal{AF}_g(\mu_{i,g}, x^{([M])})) \right) - \kappa^{(j)T} x^{(j)} \tag{87}$$

$$= \left( \sum_{i=1}^N \sum_{g \in G} p_g \mathbf{1}[\mu_{i,g} \geq p_g] \frac{1}{M} \mathcal{G}_{i,g}(x^{([M])}) \right) - \kappa^{(j)T} x^{(j)} \tag{88}$$

$$= \left( \sum_{i=1}^N \sum_{g \in G} p_g \mathbf{1}[\mu_{i,g} \geq p_g] \frac{1}{M} \mathcal{G}(x^{([M])}) \right) - \kappa^T x^{(j)} \tag{89}$$

$$= \left( \sum_{i=1}^N \sum_{g \in G} p_g \mathbf{1}[\mu_{i,g} \geq p_g] \frac{1}{M} \mathcal{G}(x_g^{([M])}) \right) - \kappa^T x^{(j)} \tag{90}$$

$$= \frac{1}{M} \left( \sum_{g \in G} p_g \left( \sum_{i=1}^N \mathbf{1}[\mu_{i,g} \geq p_g] \right) \mathcal{G}(x_g^{([M])}) \right) - \kappa^T x^{(j)} \tag{91}$$

$$= \frac{1}{M} \left( \sum_{g \in G} \rho_g \mathcal{G}(x_g^{([M])}) \right) - \kappa^T x^{(j)}. \tag{92}$$

In order: Equation (85) is by definition of the seller's utility; Equation (86) is by definition of the payment division function; Equation (87) is by Fact (A.3) since the sellers all play the same strategy; Equation (88) is by definition of the allocation function; Equation (89) is by quasi-symmetry; Equation (90) is by the assumption of zero group inter-transfer; Equation (91) is by rearranging terms; and Equation (92) is by definition of market value-of-accuracy.

In the intervention scenario, the marketplace's target vector, $\gamma$, determines the sellers' marginal production costs as $\kappa^T \gamma$, and each participating seller $j$ has only to decide how many samples to produce, $n^{(j)}$, incurring total production costs of $\kappa^T \gamma n^{(j)}$. And again, because all the seller's produce the same number of samples

at equilibrium, seller $j$'s utility becomes

$$v_j(n^{(j)}) = \frac{1}{M} \left( \sum_{g \in G} \rho_g \mathcal{G}(M\gamma_g n^{(j)}) \right) - \kappa^T \gamma n^{(j)}. \tag{93}$$

By assumption, the sellers produce some samples, i.e., $n^{([M])} > 0$, and so $n^{(j)} > 0$. It follows that there must exist at least one group $g \in G$ satisfying

$$M\gamma_h n^{(j)} > \left( \frac{\alpha}{Z} \right)^{\frac{1}{\beta}}, \tag{94}$$

because otherwise seller $j$'s utility would be negative and $\sigma$ would not be a Nash equilibrium.

Now, let $h$ be the group with the smallest $\gamma_h$ satisfying Inequality (94). Then, for every $g \in G$ satisfying $\gamma_g \geq \gamma_h$ we have

$$M\gamma_g n^{(j)} \geq M\gamma_h n^{(j)} > \left( \frac{\alpha}{Z} \right)^{\frac{1}{\beta}}. \tag{95}$$

Therefore we can define

$$H \triangleq \{g \in G : \gamma_g \geq \gamma_h\}, \tag{96}$$

and seller $j$'s utility becomes

$$v_j(n^{(j)}) = \frac{1}{M} \left( \sum_{g \in H} \rho_g \left( Z - \alpha(M\gamma_g n^{(j)})^{-\beta} \right) \right) - \kappa^T \gamma n^{(j)}. \tag{97}$$

Therefore, seller $j$'s marginal utility in $n^{(j)}$ is

$$v'(n^{(j)}) = \frac{1}{M} \sum_{g \in H} \rho_g \alpha\beta \left( M\gamma_g n^{(j)} \right)^{-\beta-1} M\gamma_g - \kappa^T \gamma. \tag{98}$$

At equilibrium, the seller's marginal utility is 0; solving for $n^{(j)}$ completes the proof. $\square$

## A.8 Proof of Claim (4.2)

**Claim 4.2.** *(Sellers' Intervention Participation Thresholds) Fix a target vector $\gamma$. Let $\sigma = (p, \{\mu_i\}, \{x^{(j)}\})$ be a Nash equilibrium. If the sellers produce a positive number of samples, i.e., $n^{([M])} > 0$, then there exists a group $h \in G$ such that the sellers' marginal production cost, $\kappa^T \gamma$, is at most a threshold value, $\tau_H(\rho, \gamma)$, given by,*

$$\tau_H(\rho, \gamma) = \frac{\left( \sum_{g \in H} \rho_g \right)^{\frac{\beta+1}{\beta}}}{\left( \sum_{g \in H} \rho_g \gamma_g^{-\beta} \right)^{\frac{1}{\beta}}} \cdot c_{\mathcal{G}} \tag{26}$$

*where $H \triangleq \{g \in G : \gamma_g \geq \gamma_h\}$,* $\tag{27}$

*and $\gamma_h$ is a minimum value over $\gamma_g$ satisfying*

$$\gamma_g n^{([M])} > \left( \frac{\alpha}{Z} \right)^{\frac{1}{\beta}}. \tag{28}$$

Proof. Fix a seller $j$, and write its utility as follows

$$v_j(x^{(j)}) = \left( \sum_{i=1}^{N} \sum_{g \in G} \mathcal{PD}_{i,g,j}(x^{(j)}) \right) - \kappa^{(j)T} x^{(j)} \tag{99}$$

$$= \sum_{i=1}^{N} \sum_{g \in G} p_g \sum_{T \subseteq [M] \setminus \{j\}} c_T \cdot \mathcal{G}_{i,g}(\mathcal{AF}_g(\mu_{i,g}, x^{(T \cup \{j\})}))$$

$$- \mathcal{G}_{i,g}(\mathcal{AF}_g(\mu_{i,g}, x^{(T)})) - \kappa^{(j)T} x^{(j)} \tag{100}$$

$$= \left( \sum_{i=1}^{N} \sum_{g \in G} p_g \frac{1}{M} \mathcal{G}_{i,g}(\mathcal{AF}_g(\mu_{i,g}, x^{([M])})) \right) - \kappa^{(j)T} x^{(j)} \tag{101}$$

$$= \left( \sum_{i=1}^{N} \sum_{g \in G} p_g \mathbf{1}[\mu_{i,g} \geq p_g] \frac{1}{M} \mathcal{G}_{i,g}(x^{([M])}) \right) - \kappa^{(j)T} x^{(j)} \tag{102}$$

$$= \left( \sum_{i=1}^{N} \sum_{g \in G} p_g \mathbf{1}[\mu_{i,g} \geq p_g] \frac{1}{M} \mathcal{G}(x^{([M])}) \right) - \kappa^{T} x^{(j)} \tag{103}$$

$$= \left( \sum_{i=1}^{N} \sum_{g \in G} p_g \mathbf{1}[\mu_{i,g} \geq p_g] \frac{1}{M} \mathcal{G}(x_g^{([M])}) \right) - \kappa^{T} x^{(j)} \tag{104}$$

$$= \frac{1}{M} \left( \sum_{g \in G} p_g \left( \sum_{i=1}^{N} \mathbf{1}[\mu_{i,g} \geq p_g] \right) \mathcal{G}(x_g^{([M])}) \right) - \kappa^{T} x^{(j)} \tag{105}$$

$$= \frac{1}{M} \left( \sum_{g \in G} \rho_g \mathcal{G}(x_g^{([M])}) \right) - \kappa^{T} x^{(j)}. \tag{106}$$

In order: Equation (99) is by definition of the seller's utility; Equation (100) is by definition of the payment division function; Equation (101) is by Fact (A.3) since the sellers all play the same strategy; Equation (102) is by definition of the allocation function; Equation (103) is by quasi-symmetry; Equation (104) is by the assumption of zero group inter-transfer; Equation (105) is by rearranging terms; and Equation (106) is by definition of economic potential.

In the intervention scenario, the marketplace's target vector, $\gamma$, determines the sellers' marginal production costs as $\kappa^T \gamma$, and each participating seller $j$ has only to decide how many samples to produce, $n^{(j)}$, incurring total production costs of $\kappa^T \gamma n^{(j)}$. And again, because all the seller's produce the same number of samples at equilibrium, seller $j$'s utility becomes

$$v_j(n^{(j)}) = \frac{1}{M} \left( \sum_{g \in G} \rho_g \mathcal{G}(M \gamma_g n^{(j)}) \right) - \kappa^T \gamma n^{(j)}. \tag{107}$$

By Lemma (4.2), there exists a group $h \in G$ such that every seller $j$ produces

$$n^{(j)} = \frac{1}{M} \left( \frac{\alpha \beta}{\kappa^T \gamma} \sum_{g \in H} \rho_g \gamma_g^{-\beta} \right)^{1/(\beta+1)} \tag{108}$$

samples, where

$$H \triangleq \{g \in G : \gamma_g \geq \gamma_h\}, \tag{109}$$

and $\gamma_h$ is a minimum value over $\gamma_g$ satisfying

$$M \gamma_g n^{(j)} > \left( \frac{\alpha}{Z} \right)^{\frac{1}{\beta}}. \tag{110}$$

Therefore, seller $j$'s utility is

$$v_j(n^{(j)}) = \frac{1}{M} \left( \sum_{g \in H} \rho_g \mathcal{G}(M \gamma_g n^{(j)}) \right) - \kappa^T \gamma n^{(j)} \tag{111}$$

$$= \frac{1}{M} \left( \sum_{g \in H} \rho_g \left( Z - \alpha \left( M \gamma_g n^{(j)} \right)^{-\beta} \right) \right) - \kappa^T \gamma n^{(j)} \tag{112}$$

$$= \frac{1}{M} \left( \sum_{g \in H} \rho_g \left( Z - \alpha \left( M \gamma_g \frac{1}{M} \left( \frac{\alpha \beta}{\kappa^T \gamma} \sum_{f \in H} \rho_f \gamma_f^{-\beta} \right)^{\frac{1}{\beta+1}} \right)^{-\beta} \right) \right)$$

$$- \kappa^T \gamma \frac{1}{M} \left( \frac{\alpha \beta}{\kappa^T \gamma} \sum_{f \in H} \rho_f \gamma_f^{-\beta} \right)^{\frac{1}{\beta+1}} \tag{113}$$

By assumption, $\sigma$ is a Nash equilibrium, hence seller $j$'s utility is non-negative; solving for $\kappa^T \gamma$ with straightforward algebra completes the proof. □

## A.9 Proof of Theorem (5.1)

**Theorem 5.1.** *For every target vector $\gamma$ there exists a data market in which $\gamma$ backfires.*

Proof. Fix a target vector $\gamma$. We will construct a data market in which $\gamma$ backfires. Let the $N$ buyers be arbitrary. Let there be $M$ sellers whose common cost structure $\kappa$ is arbitrary except for two groups $h$ and $h'$ that we will specify. We will set $\kappa_h$ so that the $h$-specific sub-market forms in the baseline scenario, and we will set $\kappa_{h'}$ so that the market does not form in the intervention scenario. In other words, we will show that: 1) in the baseline scenario there exists a Nash equilibrium $\sigma = (p, \{\mu_i\}, \{x^{(j)}\})$ such that $x_h^{([M])} > 0$; and 2) in the intervention scenario there does not exist a Nash equilibrium $\sigma' = (p, \{\mu_i\}, \{y^{(j)}\})$ such that $\|y^{([M])}\| > 0$.

We show 1) first. By Fact (4.2), the marketplace sets the reserve prices $p_g$ to maximize $\rho_g$ for every group $g$ in both scenarios. Now set $\kappa_h$ to any value satisfying $\kappa_h \leq \tau_h$. We will set $\kappa_{h'}$ more precisely when we turn to the intervention scenario, but we require it to satisfy $\kappa_{h'} > \tau_{h'}$. For every seller $j$, and group $g$ set $x_g^{(j)}$ in accordance with Theorem (4.1). It follows that $\sigma = (p, \{\mu_i\}, \{x^{(j)}\})$ is a Nash equilibrium in the baseline scenario such that $x_h^{([M])} > 0$. We conclude that the data market forms in the baseline scenario.

We now show 2). We must further specify $\kappa_{h'}$. We wish to ensure that $\kappa^T \gamma > \tau_H(\rho, \gamma)$ for every $H \subseteq G$. Observe that once the buyers are fixed, then the maximum value of $\tau_H(\rho, \gamma)$ over all possible $H$ is determined and finite. Define

$$\lceil \tau \rceil \triangleq \max_{H \subseteq G} \tau_H(\rho, \gamma). \tag{114}$$

Now we just need to ensure that

$$\kappa^T \gamma > \lceil \tau \rceil, \tag{115}$$

which readily follows by setting $\kappa_{h'}$ to any value satisfying,

$$\kappa_{h'} > \frac{\lceil \tau \rceil}{\gamma_{h'}}. \tag{116}$$

Thus, the sellers' intervention marginal cost of production is greater than intervention participation threshold over all possible $H$ and 2) follows. We conclude that the data market does not form in the intervention scenario. □

## A.10 Proof of Theorem (5.2)

**Theorem 5.2.** *Let $N$ buyers and $M$ sellers be a fully-forming data market. If the marketplace chooses the uniform intervention, i.e., $\gamma = u$, then the data market forms in the intervention scenario.*

PROOF. Fix $N$ buyers and $M$ sellers such that every group-specific sub-market forms at the Nash equilibrium, $(p, \{\mu_i\}, \{x^{(j)}\})$, in the baseline scenario. Because the market fully forms in the baseline scenario, the groups' production costs must not be too large. In particular, for each group $g$, its production costs, $\kappa_g$, must be at most the seller's baseline participation threshold, $\tau_g$,

$$\kappa_g \leq \tau_g. \tag{117}$$

We want to show that the market forms in the intervention scenario, i.e., the sellers will produce data when the marketplace applies its fairness intervention with target vector $u$. The market forms when the seller's intervention production cost, $\kappa^T u$, is no more than the seller's intervention participation threshold, $\tau$. The seller's intervention production cost is a function of $\kappa$ and $u$, hence, it is fixed once the sellers are fixed. In contrast, the seller's intervention participation threshold, $\tau$, depends on the equilibrium strategies of the marketplace and the buyers in the intervention scenario.

Let $(p, \{\mu_i\}, \{y^{(j)}\})$ be a Nash equilibrium in the intervention scenario. We must show that there exists some subset of groups $H$ such that $\kappa^T u \leq \tau_H(\rho, u)$ and for every $g \in H$, $n^{([M])}/M > (\alpha/Z)^{1/\beta}$. In fact, we will show that this holds for $H = G$. First we analyze $\tau_G(\rho, u)$,

$$\tau_G(\rho, u) = \frac{\left( \sum_{g \in G} \rho_g \right)^{\frac{\beta+1}{\beta}}}{\left( \sum_{g \in G} \rho_g \gamma_g^{-\beta} \right)^{\frac{1}{\beta}}} c_{\mathcal{G}} \tag{118}$$

$$= \frac{\left( \sum_{g \in G} \rho_g \right)^{\frac{\beta+1}{\beta}}}{\left( \sum_{g \in G} \rho_g \left( \frac{1}{|G|} \right)^{-\beta} \right)^{\frac{1}{\beta}}} c_{\mathcal{G}} \tag{119}$$

$$= \frac{\left( \sum_{g \in G} \rho_g \right)^{\frac{\beta+1}{\beta}}}{\left( \sum_{g \in G} \rho_g |G|^{\beta} \right)^{\frac{1}{\beta}}} c_{\mathcal{G}} \tag{120}$$

$$= \frac{\left( \sum_{g \in G} \rho_g \right)^{\frac{\beta+1}{\beta}}}{\left( |G|^{\beta} \sum_{g \in G} \rho_g \right)^{\frac{1}{\beta}}} c_{\mathcal{G}} \tag{121}$$

$$= \frac{\left( \sum_{g \in G} \rho_g \right)^{\frac{\beta+1}{\beta}}}{|G| \left( \sum_{g \in G} \rho_g \right)^{\frac{1}{\beta}}} c_{\mathcal{G}} \tag{122}$$

$$= \frac{1}{|G|} \sum_{g \in G} \rho_g c_{\mathcal{G}} \tag{123}$$

$$= \frac{1}{|G|} \sum_{g \in G} \tau_g. \tag{124}$$

By assumption, $\kappa_g \leq \tau_g$ for every group $g$. It follows that

$$\kappa^T u = \frac{1}{|G|} \sum_{g \in G} \kappa_g \leq \frac{1}{|G|} \sum_{g \in G} \tau_g = \tau_G(\rho, u). \tag{125}$$

It remains to show that

$$y_g^{([M])} = \frac{n^{([M])}}{M} > \left( \frac{\alpha}{Z} \right)^{\frac{1}{\beta}}. \tag{126}$$

When $H = G$ we have,

$$n^{([M])} = \left( \frac{\alpha \beta}{\kappa^T u} \sum_{g \in G} \rho_g u_g^{-\beta} \right)^{\frac{1}{\beta+1}} \tag{127}$$

$$= \left( \frac{\alpha \beta}{\sum_{g \in G} \kappa \frac{1}{M}} \sum_{g \in G} \rho_g \left( \frac{1}{M} \right)^{-\beta} \right)^{\frac{1}{\beta+1}} \tag{128}$$

$$= M \left( \alpha \beta \frac{\sum_{g \in G} \rho_g}{\sum_{g \in G} \kappa_g} \right)^{\frac{1}{\beta+1}} \tag{129}$$

$$\geq M \left( \frac{\alpha \beta}{c_{\mathcal{G}}} \right) > M \left( \frac{\alpha}{Z} \right)^{\frac{1}{\beta}}. \tag{130}$$

This completes the proof. □

## A.11 Proof of Theorem (5.3)

**Theorem 5.3.** *Let $\gamma$ be the marketplace's target vector. If $\gamma$ is not the uniform intervention, i.e., $\gamma \neq u$, then there exists a data-market*

*that is fully forming in the baseline scenario but does not form in the intervention scenario.*

Let us first discuss the proof. The key high-level idea is the following. We will construct a specific data market in which every $\gamma \neq u$ backfires.[3] This will be due to two key features of the data market: 1) it is just on the verge of formation; and 2) the dataset demographics at baseline equilibrium will be uniform. Thus, any $\gamma$ that forces the sellers to deviate from their baseline equilibrium strategies also forces the sellers out of the data market. Although the result is intuitive, the proof is quite involved due to one principal technical challenge.

For any $\gamma \neq u$, it is straightforward to check that if the marketplace sets the same reserve prices in the intervention scenario as in the baseline-scenario equilibrium, then the market will not form in the intervention scenario. This observation proves that $\gamma$ backfires when the marketplace does not change its reserve prices. The technical difficulty stems from the fact that the marketplace can, in principle, set different reserve prices in the intervention scenario. And Claim (4.2) indicates that sometimes the marketplace can increase the sellers' intervention participation threshold, $\tau$, by *decreasing* some of its reserve prices. To prove the theorem requires us to prove that $\gamma$ backfires for any set of reserve prices the marketplace may choose, not just the same ones as at baseline equilibrium. And this must be over all possible choices of $\gamma$.

The proof establishes the theorem by showing that over all possible $\gamma$ and all possible reserve prices, the sellers' intervention participation threshold is less than the seller's marginal cost of production when $\gamma \neq u$. This is done in two major steps. In the first step, the quantification over all possible $\gamma$ and reserve prices is reduced to a quantification over all possible reserve prices by computing the $\gamma$ that maximizes the sellers' intervention participation threshold given a fixed set of reserve prices. And the second step proves that the maximum sellers' intervention participation threshold over all reserve prices is obtained when $\gamma = u$. This proves the theorem, because that is precisely the seller's marginal cost of production.

Proof. We first specify the data market. There is only 1 buyer whose value-of-accuracy is the same across all the groups, i.e., for all $g \in G$, we have $\mu_{1,g} = c_\mu$ for some positive constant $c_\mu > 0$. There are $M$ sellers that face a cost-structure $\kappa$, with the following two properties: 1) the marginal production cost is the same across all the groups, i.e., for all $g \in G$, we have $\kappa_g = c_\kappa$ for some positive constant $c_\kappa > 0$; and 2) $c_\kappa$ is related to the buyer's value-of-accuracy and prediction gain function by $c_\kappa = c_\mu c_{\mathcal{G}}$.

We now analyze the baseline equilibrium. Since there is only one buyer, the marketplace's value-of-accuracy for group $g$ when it plays reserve price $p_g$ is $p_g$ if $p_g \leq c_\mu$ and 0 otherwise. Therefore, the marketplace will set each group's reserve price to the buyer's value-of-accuracy and all the reserve prices will be the same, i.e., for all $g \in G$, $p_g = c_\mu$. Consequently, the seller's baseline participation threshold for each group $g$ is $\tau_g = c_\mu c_{\mathcal{G}}$. By construction, for every group $g$, $\kappa_g = c_\kappa = c_\mu c_{\mathcal{G}}$, therefore $\kappa_g = \tau_g$ and every seller will produce samples of group $g$ at equilibrium, i.e., $x_g^{(j)} > 0$. Moreover, every seller will produce the same number of samples for every

group, i.e., for all $g, h \in G$, $x_g^{(j)} = x_h^{(j)}$. Therefore every group will have the same number of samples in the aggregate dataset, i.e., for all $g, h \in G$, $x_g^{([M])} = x_h^{([M])}$, and the demographics of the aggregate dataset will coincide with the uniform intervention, i.e., $\gamma(x^{([M])}) = u$.

We turn to analyzing the intervention scenario. We show that over all the marketplace's possible choices of target vector, $\gamma$, and reserve prices $q$, the sellers' intervention participation threshold reaches its maximum when $\gamma = u$ and $q_g = c_\mu$. We do this in two major steps. In the first step, we solve for the target vector $\gamma$ that maximizes the seller's intervention participation threshold given fixed reserve prices $q$. This allows us to maximize the sellers' intervention participation threshold solely in terms of the reserve prices. In the second step, we show that the reserve prices that maximize the seller's intervention participation threshold are $q_g = c_\mu$, and this implies that $\gamma = u$.

We now take the first step. Fix the marketplace's reserve prices $q$ in the intervention scenario. The sellers' intervention participation threshold is

$$\tau = \frac{\left(\sum_{g \in H} \rho_g\right)^{\frac{\beta+1}{\beta}}}{\left(\sum_{g \in H} \rho_g \gamma_g^{-\beta}\right)^{\frac{1}{\beta}}} \cdot c_{\mathcal{G}}. \tag{131}$$

Note that $\tau$ depends on $q$ through $\rho_g$, and that the $\rho_g$ are fixed because $q$ is fixed. Therefore, choosing $\gamma$ to maximize $\tau$ is equivalent to choosing $\gamma$ to minimize

$$\sum_{g \in H} \rho_g \gamma_g^{-\beta}. \tag{132}$$

Define

$$f(\gamma) \triangleq \begin{cases} \sum_{g \in H} \rho_g \gamma_g^{-\beta} & \text{if } \forall h \in H, \gamma_h > 0 \\ \infty & \text{otherwise} \end{cases} \tag{133}$$

Thus, we wish to solve the following program:

$$\min_\gamma f(\gamma), \tag{134}$$

subject to

$$\sum_{g \in G} \gamma_g = 1, \tag{135}$$

and for all $g \in G$, $0 \leq \gamma_g \leq 1$.

Define the following functions for the constraints:

$$h(\gamma) \triangleq \sum_{g \in G} \gamma_g - 1; \tag{136}$$

and for every $g \in G$,

$$b_{(\ell,g)}(\gamma) \triangleq -\gamma_g, \tag{137}$$

and

$$b_{(u,g)}(\gamma) \triangleq \gamma_g - 1. \tag{138}$$

Compute the partial derivatives of the objective and constraint functions. For the objective,

$$\frac{\partial}{\partial \gamma_g} f(\gamma) = \begin{cases} -\beta \rho_g \gamma_g^{-\beta-1} & \text{if } g \in H \\ 0 & \text{otherwise} \end{cases} \tag{139}$$

For the equality constraint,

$$\frac{\partial}{\partial \gamma_g} h(\gamma) = 1. \tag{140}$$

---

[3]We analyze a specific data market for ease and clarity of presentation, our analysis readily generalizes to a more restricted class.

For the lower bound constraints,

$$\frac{\partial}{\partial \gamma_h} b_{(\ell,g)}(\gamma) = \begin{cases} -1 & \text{if } g = h \\ 0 & \text{otherwise} \end{cases} \tag{141}$$

And for the upper bound constraints,

$$\frac{\partial}{\partial \gamma_h} b_{(u,g)}(\gamma) = \begin{cases} 1 & \text{if } g = h \\ 0 & \text{otherwise} \end{cases} \tag{142}$$

By the Karush-Kuhn-Tucker (KKT) conditions, we are searching for solutions $\gamma$ that satisfy the multiplier rule,

$$\nabla f(\gamma) + \nabla b(\gamma)\lambda + \nabla h(\gamma)\mu = 0, \tag{143}$$

and complementarity conditions, i.e., $\lambda \geq 0$ and

$$b(\gamma)^T \lambda = 0. \tag{144}$$

Plug the partial derivatives into the KKT multiplier rule. For each $g \in H$ this gives

$$-\beta \rho_g \gamma_g^{-\beta-1} - \lambda_{(\ell,g)} + \lambda_{(u,g)} + \mu = 0. \tag{145}$$

And for each $g' \in G \setminus H$ this gives

$$-\lambda_{(\ell,g')} + \lambda_{(u,g')} + \mu = 0. \tag{146}$$

Now analyze the multipliers. By the definition of $f$, observe that an optimal solution $\gamma$ must satisfy $\gamma_g > 0$ for all $g \in H$. This has a number of consequences. First, if $g \in H$, then the lower bound constraint is loose, i.e., $b_{(\ell,g)}(\gamma) < 0$, and the complementarity conditions imply that $\lambda_{(\ell,g)} = 0$. Second, if $g' \in G \setminus H$, then the upper bound constraint for $g'$ is loose, i.e., $\gamma_{g'} < 1$, and the complementarity conditions imply that $\lambda_{(u,g')} = 0$. Finally, if $g' \in G \setminus H$, then $\gamma_{g'} = 0$, which can be seen as follows. Towards contradiction, suppose $\gamma_{g'} > 0$. Then the KKT multiplier rule for $g'$ is

$$\mu = 0, \tag{147}$$

and the KKT multiplier rule for any $g \in H$ is

$$-\beta \rho_g \gamma_g^{-\beta-1} + \mu = -\beta \rho_g \gamma_g^{-\beta-1} + 0 = -\beta \rho_g \gamma_g^{-\beta-1} = 0. \tag{148}$$

But this is a contradiction because $\gamma_g > 0$ implies that $-\beta \rho_g \gamma_g^{-\beta-1} < 0$.

It remains to solve for $\gamma_g$, for each $g \in H$. If $|H| = 1$, it follows that $\gamma_g = 1$. Otherwise, $|H| > 1$, and the KKT multiplier rule is

$$-\beta \rho_g \gamma_g^{-\beta+1} + \mu = 0. \tag{149}$$

It follows that for every $h \neq g \in H$ we have

$$-\beta \rho_g \gamma_g^{-\beta+1} = -\beta \rho_h \gamma_h^{-\beta+1}, \tag{150}$$

and with some straightforward algebra we obtain

$$\gamma_h = \left(\frac{\rho_h}{\rho_g}\right)^{\frac{1}{\beta+1}} \gamma_g. \tag{151}$$

Plugging this into the equality constraint we obtain

$$\sum_{h \in H} \gamma_h = \sum_{h \in H} \left(\frac{\rho_h}{\rho_g}\right)^{\frac{1}{\beta+1}} \gamma_g = 1. \tag{152}$$

And solving for $\gamma_g$ yields

$$\gamma_g = \frac{\rho_g^{\frac{1}{\beta+1}}}{\sum_{h \in H} \rho_h^{\frac{1}{\beta+1}}}. \tag{153}$$

Plugging the solution for $\gamma$ into the sellers' intervention participation threshold $\tau$, with some straightforward algebra, we obtain that the maximum $\tau$ can be over all choices of $\gamma$ for a fixed set of reserve prices is

$$\tau = \left(\frac{\sum_{h \in H} \rho_h}{\sum_{h \in H} \rho_h^{\frac{1}{\beta+1}}}\right)^{\frac{\beta+1}{\beta}} c_{\mathcal{G}}. \tag{154}$$

This completes the first major step.

We move on to the second step. What is the maximum sellers' intervention participaiton threshold, $\tau$, over all the possible reserve prices, $q$, the marketplace can set? We first derive a non-standard program and then formulate an equivalent program in standard form. First, what are the possible reserve prices the marketplace can set? In principle, the marketplace has the flexibility to set $p_g$ to any non-negative value for each group $g \in G$, i.e., $p_g \geq 0$. Moreover $p_g$ enters $\tau$ via $\rho_g = p_g \mathbf{1}[b_{1,g} \geq p_g]$. At intervention equilibrium, the buyer will bid its value-of-accuracy for each group $g \in G$, $\mu_{1,g} = c_\mu$. Therefore, the marketplace can set $\rho_g$ to any value in $[0, c_\mu]$ by setting $p_g$ appropriately. Thus, choosing $p_g$ is equivalent to choosing $\rho_g$, so we now focus on the marketplace's choices of $\rho$.

In this problem, $\rho$ determines $\gamma$. $\gamma$ is set to maximize the sellers' intervention participation threshold. As we have just shown, this entails that any unmonetized group has no representation, i.e., for any $g \in G \setminus H$, $\rho_g = 0$. Thus the search space over $\rho$ is 0 if $g in G \setminus H$ and $[0, c_\mu$ if $g \in H$. We have derived the following program:

$$\max_\rho \left(\frac{\sum_{h \in H} \rho_h}{\sum_{h \in H} \rho_h^{\frac{1}{\beta+1}}}\right)^{\frac{\beta+1}{\beta}}, \tag{155}$$

subject to

$$0 \leq \rho_g \leq c_\mu, \tag{156}$$

for all $g \in H$, where

$$\rho_g = 0, \tag{157}$$

for all $g \in G \setminus H$.

We now formulate an equivalent program in standard form. Define objective,

$$f(\rho) \triangleq -\frac{\sum_{h \in H} \rho_h}{\sum_{h \in H} \rho_h^{\frac{1}{\beta+1}}}. \tag{158}$$

Define equality constraints: For each $g \in G \setminus H$,

$$h_g(\rho) = \rho_g, \tag{159}$$

subject to,

$$h_g(\rho) = 0. \tag{160}$$

Define inequality constraints: For each $g \in H$,

$$b_{(\ell,g)}(\rho) = -\rho_g \tag{161}$$

and

$$b_{(u,g)}(\rho) = \rho_g - c_\mu. \tag{162}$$

And define the program

$$\min_\rho f(\rho), \tag{163}$$

subject to

$$b(\rho) \le 0. \tag{164}$$

We now solve the program. For each $g \in G \setminus H$, the equality constraint $h_g(\rho)$ determines $\rho_g = 0$. We turn to solving $\rho_g$ for $g \in H$. Compute the partial derivatives of the objective and inequality constraint functions. For the objective,

$$\frac{\partial}{\partial \rho_g} f(\rho) = -\frac{\left(\sum_{h \in H} \rho_g^{\frac{1}{\beta+1}}\right) - \left(\sum_{h \in H} \rho_h\right) \frac{1}{\beta+1} \rho_g^{-\frac{\beta}{\beta+1}}}{\left(\sum_{h \in H} \rho_h^{\frac{1}{\beta+1}}\right)^2} \tag{165}$$

For the lower bound constraints,

$$\frac{\partial}{\partial \rho_h} b_{(\ell,g)}(\rho) = \begin{cases} -1 & \text{if } h = g \\ 0 & \text{otherwise} \end{cases} \tag{166}$$

And for the upper bound constraints,

$$\frac{\partial}{\partial \rho_h} b_{(u,g)}(\rho) = \begin{cases} 1 & \text{if } h = g \\ 0 & \text{otherwise} \end{cases} \tag{167}$$

We first find the feasible solutions that satisfy the KKT conditions. The multiplier rule of the KKT conditions gives

$$\frac{\partial}{\partial \rho_g} f(\rho) + \frac{\partial}{\partial \rho_g} b_{(\ell,g)}(\rho) \lambda_{(\ell,g)} + \frac{\partial}{\partial \rho_g} b_{(u,g)}(\rho) \lambda_{(u,g)} = 0. \tag{168}$$

Substituting the partial derivatives of the inequality constraints yields,

$$\frac{\partial}{\partial \rho_g} f(\rho) - \lambda_{(\ell,g)} + \lambda_{(u,g)} = 0. \tag{169}$$

Observe that $\rho = 0$ is not a local minimum. And for every feasible solution $\rho \ne 0$ we have

$$\frac{\partial}{\partial \rho_g} f(\rho) \ne 0. \tag{170}$$

It follows that if a feasible solution $\rho$ satisfies the KKT conditions, then for all $g \in H$ either $\rho_g = 0$ or $\rho_g = c_\mu$. Let us call such a feasible solution a corner. Moreover, every corner satisfies the KKT conditions by setting their multipliers appropriately. It follows that if there is a local minimum, it is one of these corners.

We now show that every corner $\rho$ is a local minimum. It suffices to show that for every $g \in H$ the following two implications hold: 1) if $\rho_g = 0$, then $\frac{\partial}{\partial \rho_g} f(\rho) > 0$; and 2) if $\rho_g = c_\mu$, then $\frac{\partial}{\partial \rho_g} f(\rho) < 0$. And observe that in both cases, we have

$$\frac{\partial}{\partial \rho_g} f(\rho) = -\frac{c_\mu^{\frac{1}{\beta+1}} - \frac{1}{\beta+1} c_\mu \rho_g^{-\frac{\beta}{\beta+1}}}{n(c_\mu^{\frac{1}{\beta+1}})^2}, \tag{171}$$

where $n$ is the number of groups $h$ such that $\rho_h = c_\mu$, because $\rho$ is a corner.

Now consider the first implication. Suppose $\rho_g = 0$. Observe that $\rho_g^{-\frac{\beta}{\beta+1}} \to \infty$ as $\rho_g \to 0$. We conclude that $\frac{\partial}{\partial \rho_g} f(\rho) > 0$.

Now consider the second implication. Suppose $\rho_g = c_\mu$. Then,

$$\frac{\partial}{\partial \rho_g} f(\rho) = -\frac{\beta}{(\beta+1)n} < 0. \tag{172}$$

Therefore every corner $\rho$ is a local minimum. Actually, every corner is a global minimum because since they all achieve the same objective value,

$$f(\rho) = -\frac{\sum_{h \in H} \rho_h}{\sum_{h \in H} \rho_h^{\frac{1}{\beta+1}}} = -\frac{nc_\mu}{nc_\mu^{\frac{1}{\beta+1}}} = c_\mu^{\frac{\beta}{\beta+1}}, \tag{173}$$

where $n$ is the number of groups $h$ such that $\rho_h = c_\mu$.

We conclude that the maximum sellers' intervention participation threshold is achieved when the intervention stipulates uniform intervention over the monetized groups and the marketplace sets the reserve prices as high as the buyers can bear. Since the marketplace wishes to ensure that there is representation for all the groups in the data market, this implies that the only such feasible intervention is the uniform intervention, i.e., $\gamma = u$. When the intervention is $u$, the sellers' intervention participation threshold becomes,

$$\tau(u) = \left(\frac{\sum_{h \in H} \rho_h}{\sum_{h \in H} \rho_h^{\frac{1}{\beta+1}}}\right)^{\frac{\beta+1}{\beta}} c_G = c_\mu c_\kappa = \kappa^T u, \tag{174}$$

and the market will still form in the intervention scenario. But for any other feasible $\gamma \ne u$, this implies that

$$\tau(u) = \left(\frac{\sum_{h \in H} \rho_h}{\sum_{h \in H} \rho_h^{\frac{1}{\beta+1}}}\right)^{\frac{\beta+1}{\beta}} c_G < c_\mu c_\kappa = \kappa^T \gamma, \tag{175}$$

and the market will not form in the intervention scenario. This concludes the proof. □

## A.12 Proof of Theorem (5.4)

**Theorem 5.4.** *Let N buyers and M sellers be a fully-forming data market. Define $\eta \in [0, 1]$ to be the minimum value satisfying for all $g \in G$,*

$$\kappa_g \le \eta \tau_g. \tag{32}$$

*Let $\gamma$ be an intervention. Define $a \ge 1$,*

$$\frac{1}{a} = \min_{g \in G} \gamma_g, \tag{33}$$

*and $b \ge 1$*

$$\frac{1}{b} = \max_{g \in G} \gamma_g. \tag{34}$$

*If the marketplace chooses target vector $\gamma$ and*

$$\eta < \left(\frac{b}{a}\right)^{\beta+1} \frac{1}{r|G|}, \tag{35}$$

*where $r$ is a constant that depends on the N buyers, then the market will form in the intervention scenario.*

Proof. We will show that there must exist a subset of groups that can be profitably monetized in the intervention scenario, i.e., there exists a strategy profile and a subset of groups such that in the intervention scenario: 1) the sellers' intervention marginal

production cost is at most the sellers' intervention participation threshold; and 2) the sellers produce more than the learning ante for every group in the subset. This implies that the market will form in the intervention scenario.

We first show that the sellers' intervention marginal production cost is at most the sellers' intervention participation threshold. Since $\mathcal{M}$ is a fully-forming data market in the baseline scenario, there exists a strategy profile $\sigma = (p, \{\mu_i\}, \{x^{(j)}\})$ such that $x_g^{([M])} > 0$ for all $g \in G$, and $\sigma$ is a Nash equilibrium in the baseline scenario. Consider the following strategy profile $\sigma' = (q, \{\mu_i\}, \{n^{(j)}\})$ in the intervention scenario where $q = p$ and

$$n^{(j)} = \frac{1}{M} \left( \frac{\alpha\beta}{\kappa^T \gamma} \sum_{h \in H} \rho_h \gamma_h^{-\beta} \right)^{\frac{1}{\beta}}, \tag{176}$$

for some subset of monetized groups $H \subseteq G, H \neq \emptyset$.

Observe that we can bound the sellers' intervention production costs, $\kappa^T \gamma$, from above,

$$\kappa^T \gamma = \sum_{g \in G} \kappa_g \gamma_g \leq \sum_{g \in G} \eta \tau_g \gamma_g \leq \sum_{g \in G} \eta \tau_g \frac{1}{b} = \frac{\eta}{b} \sum_{g \in G} \rho_g c_{\mathcal{G}}. \tag{177}$$

The sellers' intervention participation threshold, $\tau$, depends on the subset of monetized groups, $H$. And observe that we can bound $\tau$ from below,

$$\tau = \frac{\left( \sum_{g \in H} \rho_g \right)^{\frac{\beta+1}{\beta}}}{\left( \sum_{g \in H} \rho_g \gamma_g^{-\beta} \right)^{\frac{1}{\beta}}} c_{\mathcal{G}} = \left( \frac{\sum_{g \in H} \rho_g}{\sum_{g \in H} \rho_g \gamma_g^{-\beta}} \right)^{\frac{1}{\beta}} \left( \sum_{g \in H} \rho_g \right) c_{\mathcal{G}} \tag{178}$$

$$\geq \left( \frac{\sum_{g \in H} \rho_g}{\sum_{g \in H} \rho_g \frac{1}{a}^{-\beta}} \right)^{\frac{1}{\beta}} \sum_{g \in H} \rho_g c_{\mathcal{G}} = \frac{1}{a} \sum_{g \in H} \rho_g c_{\mathcal{G}}. \tag{179}$$

Except for their index sets, the bounds are structurally very similar. We now derive an inequality to bridge the difference in their index sets. Define $\overline{H} \triangleq G \setminus H$,

$$\rho_0 \triangleq \min_{h \in H} \rho_h, \tag{180}$$

and

$$r \triangleq \max_{f, g \in G} \frac{\rho_f}{\rho_g}, \tag{181}$$

and write

$$\sum_{g \in G} \rho_g = \sum_{h \in H} \rho_h + \sum_{g \in \overline{H}} \rho_g = \sum_{h \in H} \rho_h + \sum_{g \in \overline{H}} \frac{\rho_0}{\rho_0} \rho_g = \sum_{h \in H} \rho_h + \sum_{g \in \overline{H}} \frac{\rho_g}{\rho_0} \rho_0 \tag{182}$$

$$\leq \sum_{h \in H} \rho_h + \sum_{g \in \overline{H}} r \rho_0 = \sum_{h \in H} \rho_h + r |\overline{H}| \rho_0 \tag{183}$$

$$\leq \sum_{h \in H} \rho_h + r |\overline{H}| \left( \frac{1}{|H|} \sum_{h \in H} \rho_h \right) = \left( 1 + r \frac{|\overline{H}|}{|H|} \right) \sum_{h \in H} \rho_h. \tag{184}$$

Now $1 \leq |H| \leq |G|$, and $|\overline{H}| = |G| - |H|$ so

$$\frac{|\overline{H}|}{|H|} = \frac{|G| - |H|}{|H|} = \frac{|G|}{|H|} - 1 \leq |G| - 1, \tag{185}$$

and therefore

$$1 + r \frac{|\overline{H}|}{|H|} \leq 1 + r(|G| - 1) = 1 + r|G| - r \leq r|G|. \tag{186}$$

Putting Inequalities (184) and (186) together we have,

$$\sum_{g \in G} \rho_g \leq r|G| \sum_{h \in H} \rho_h. \tag{187}$$

Applying this to Inequality (177) we obtain,

$$\kappa^T \gamma \leq \frac{\eta}{b} r|G| \sum_{h \in H} \rho_h c_{\mathcal{G}}. \tag{188}$$

By definition, $b \leq a$, and by assumption it follows that

$$\eta < \left( \frac{b}{a} \right)^{\beta+1} \frac{1}{r|G|} < \frac{b}{a} \frac{1}{r|G|}. \tag{189}$$

Applying this to Inequality (188) yields,

$$\kappa^T \gamma \leq \frac{1}{a} \sum_{h \in H} \rho_h c_{\mathcal{G}}, \tag{190}$$

which we recognize as the lower bound on $\tau$ in Inequality (179), i.e.,

$$\kappa^T \gamma \leq \tau. \tag{191}$$

Critically, the analysis of $\kappa^T \gamma$ and $\tau$ depends on the sellers producing more than the learning ante number of samples for each group in $H$, that is, it requires,

$$\gamma_h n^{([M])} > \left( \frac{\alpha}{Z} \right)^{\frac{1}{\beta}}, \tag{192}$$

for every $h \in H$. We now show that this holds.

$$\gamma_h n^{([M])} = \gamma_h \left( \frac{\alpha\beta}{\kappa^T \gamma} \sum_{g \in H} \rho_g \gamma_g^{-\beta} \right)^{\frac{1}{\beta+1}} \tag{193}$$

$$\geq \frac{1}{a} \left( \frac{\alpha\beta}{\kappa^T \gamma} \sum_{g \in H} \rho_g \gamma_g^{-\beta} \right)^{\frac{1}{\beta+1}} \tag{194}$$

$$\geq \frac{1}{a} \left( \frac{\alpha\beta}{\kappa^T \gamma} \sum_{g \in H} \rho_g b^\beta \right)^{\frac{1}{\beta+1}} \tag{195}$$

$$\geq \frac{1}{a} \left( \alpha\beta \frac{b}{\eta r|G| \sum_{h \in H} \rho_h c_{\mathcal{G}}} \sum_{g \in H} \rho_g b^\beta \right)^{\frac{1}{\beta+1}} \tag{196}$$

$$= \frac{b}{a} \left( \frac{\alpha\beta}{\eta r|G| c_{\mathcal{G}}} \right)^{\frac{1}{\beta+1}} \tag{197}$$

$$= \frac{b}{a} \left( \frac{\alpha\beta}{\eta r|G|} \frac{\alpha^{\frac{1}{\beta}} \left( \beta^{-\frac{\beta}{\beta+1}} + \beta^{\frac{1}{\beta+1}} \right)^{\frac{\beta+1}{\beta}}}{Z^{\frac{\beta+1}{\beta}}} \right)^{\frac{1}{\beta+1}} \tag{198}$$

$$\geq \frac{b}{a} \left( \frac{1}{\eta r|G|} \left( \frac{\alpha}{Z} \right)^{\frac{\beta+1}{\beta}} \right)^{\frac{1}{\beta+1}} \tag{199}$$

$$= \frac{b}{a} \left( \frac{1}{\eta r|G|} \right)^{\frac{1}{\beta+1}} \left( \frac{\alpha}{Z} \right)^{\frac{1}{\beta}} > \left( \frac{\alpha}{Z} \right)^{\frac{1}{\beta}} \tag{200}$$

because

$$\frac{b}{a}\left(\frac{1}{\eta r |G|}\right)^{\frac{1}{\beta+1}} > 1 \tag{201}$$

by assumption. We conclude that there exists a subset of groups $H$ that can be profitably monetized. □

### A.13 Proof of Claim (6.1)

**Claim 6.1.** *Let $\gamma$ be the marketplace's target vector. If there exists $g \in G$ such that $\max_{p_g} \rho_g \to \infty$ as $N \to \infty$, then there exists an $N_0$ such that $N > N_0$ implies that for all $j$, $\|y^{(j)}\| > 0$.*

Proof. Let $\lceil \gamma \rceil \triangleq \max_{g \in G} \gamma_g^{-\beta}$. In the intervention scenario, By Claim (4.2), the sellers will produce data if

$$\kappa^T \gamma \leq \frac{\left(\sum_{g \in G} \rho_g\right)^{\frac{\beta+1}{\beta}}}{\left(\sum_{g \in G} \rho_g \gamma_g^{-\beta}\right)^{\frac{1}{\beta}}} \cdot \frac{Z^{\frac{\beta+1}{\beta}}}{\alpha^{\frac{1}{\beta}}\left(\beta^{-\frac{\beta}{\beta+1}} + \beta^{\frac{1}{\beta+1}}\right)^{\frac{\beta+1}{\beta}}}, \tag{202}$$

or equivalently

$$\frac{\alpha\left(\beta^{-\frac{\beta}{\beta+1}} + \beta^{\frac{1}{\beta+1}}\right)^{\beta+1}}{Z^{\beta+1}}\left(\kappa^T \gamma\right)^\beta \leq \frac{\left(\sum_{g \in G} \rho_g\right)^{\beta+1}}{\left(\sum_{g \in G} \rho_g \gamma_g^{-\beta}\right)} \tag{203}$$

Recall that the $\gamma_g$ are fixed with respect to increasing $N$. Bound the right-hand side of Inequality (203) from below by:

$$\frac{1}{\lceil \gamma \rceil}\left(\sum_{g \in G} \rho_g\right)^\beta \leq \frac{\left(\sum_{g \in G} \rho_g\right)^{\beta+1}}{\left(\sum_{g \in G} \rho_g \gamma_g^{-\beta}\right)} \tag{204}$$

Therefore, Inequality (203) will hold if

$$\frac{\alpha\left(\beta^{-\frac{\beta}{\beta+1}} + \beta^{\frac{1}{\beta+1}}\right)^{\beta+1}}{Z^{\beta+1}}\left(\kappa^T \gamma\right)^\beta \leq \left(\frac{1}{\lceil \gamma \rceil}\sum_{g \in G} \rho_g\right)^\beta \tag{205}$$

Recall that $Z$, $\alpha$, $\beta$, and $\kappa_g$ are fixed with respect to increasing $N$. By assumption, there is at least one group $g$ such that $\max_{p_g} \rho_g \to \infty$ as $N \to \infty$. Observe that $\max_{p_g} \rho_g$ is non-decreasing in the number of buyers $N$. Therefore there is some $N_0$ such that Inequality (205) will be satisfied for all $N > N_0$. □

### A.14 Proof of Claim (A.1)

**Claim A.1.** *If $\max_{p_g} \rho_g \to \infty$ as $N \to \infty$, then there exists an $N_0$ such that $N > N_0$ implies that for all $j$, $x_g^{(j)} > 0$.*

Proof. In the baseline scenario, the sellers will produce data if

$$\kappa_g \leq \tau_g = \frac{\rho_g Z^{\frac{\beta+1}{\beta}}}{\alpha^{\frac{1}{\beta}}\left(\beta^{-\frac{\beta}{\beta+1}} + \beta^{\frac{1}{\beta+1}}\right)^{\frac{\beta+1}{\beta}}}, \tag{206}$$

or equivalently

$$\frac{\alpha^{\frac{1}{\beta}}\left(\beta^{-\frac{\beta}{\beta+1}} + \beta^{\frac{1}{\beta+1}}\right)^{\frac{\beta+1}{\beta}}}{Z^{\frac{\beta+1}{\beta}}}\kappa_g \leq \rho_g \tag{207}$$

Recall that $Z$, $\alpha$, $\beta$, and $\kappa_g$ are fixed with respect to increasing $N$. By assumption, $\max_{p_g} \rho_g \to \infty$ as $N \to \infty$ and note that $\max_{p_g} \rho_g$ is non-decreasing in the number of buyers $N$, therefore there is some $N_0$ such that Inequality (207) is satisfied for every $N > N_0$. □

### A.15 Proof of Claim (A.2)

**Claim A.2.** *If $\max_{p_g} \rho_g \to \infty$ as $N \to \infty$, then $x_g^{([M])} \to \infty$.*

Proof. By Lemma (??), at equilibrium we have that

$$x_g^{([M])} = \left(\frac{\rho_g}{\kappa_g}\alpha\beta\right)^{1/(\beta+1)}. \tag{208}$$

By assumption, $\max_{p_g} \rho_g \to \infty$, it follows that by Claim (A.1) the sellers will participate in the market for all $N$ sufficiently large and therefore

$$x_g^{([M])} = \left(\frac{\rho_g}{\kappa_g}\alpha\beta\right)^{1/(\beta+1)} \to \infty. \tag{209}$$

□

### A.16 Proof of Claim (A.3)

**Claim A.3.** *If there exists $g \in G$ such that $\max_{p_g} \rho_g \to \infty$ as $N \to \infty$, then $n^{([M])} = \|y^{([M])}\| \to \infty$.*

Proof. By Lemma (??) and Claim (4.2) the sellers will produce

$$n^{([M])} = \left(\frac{\alpha\beta}{\kappa^T \gamma}\sum_{g \in H} \rho_g \gamma_g^{-\beta}\right)^{1/(\beta+1)} \tag{210}$$

samples at equilibrium if

$$\kappa^T \gamma \leq \tau_H(\rho, \gamma) \tag{211}$$

where

$$H \triangleq \{g \in G : \gamma_g \geq \gamma_h\}, \tag{212}$$

and $\gamma_h$ is a minimum value over $\gamma_g$ satisfying

$$\gamma_g n^{([M])} > \left(\frac{\alpha}{Z}\right)^{\frac{1}{\beta}}. \tag{213}$$

Observe that $\kappa^T \gamma$ is fixed as $\rho_g \to \infty$, whereas $n^{([M])}$ and $\tau_G(\rho, \gamma)$ grow unboundedly. □

### A.17 Proof of Theorem (6.1)

**Theorem 6.1.** *If there exists $g \in G$ such that $\max_{p_g} \rho_g \to \infty$ as $N \to \infty$, then for the marketplace we have*

$$\lim_{N \to \infty} \frac{w^f(p)}{w(p)} = 1, \tag{39}$$

*for every seller $j$ we have*

$$\lim_{N \to \infty} \frac{v_j^f(y^{(j)})}{v_j(x^{(j)})} = 1, \tag{40}$$

*and for every buyer $i$ we have*

$$\lim_{N \to \infty} \frac{u_i^f(\mu_i)}{u_i(\mu_i)} \geq 1. \tag{41}$$

Proof. Consider the marketplace's utility in the baseline scenario.

$$w(p) = \sum_{g \in G} p_g \sum_{i=1}^{N} \mathbf{1}[\mu_{i,g} \geq p_g] \mathcal{G}(x_g^{([M])}) \tag{214}$$

$$= \sum_{g \in G} \rho_g \left( Z - \alpha (x_g^{([M])})^{-\beta} \right) \tag{215}$$

$$= Z \sum_{g \in G} \rho_g - \alpha \sum_{g \in G} \rho_g (x_g^{([M])})^{-\beta} \tag{216}$$

$$= Z \sum_{g \in G} \rho_g - \alpha \sum_{g \in G} \rho_g \left( \frac{\rho_g}{\kappa_g} \alpha \beta \right)^{-\frac{\beta}{\beta+1}} \tag{217}$$

$$= Z \sum_{g \in G} \rho_g - \alpha \sum_{g \in G} \rho_g^{\frac{1}{\beta+1}} \left( \frac{\alpha \beta}{\kappa_g} \right)^{-\frac{\beta}{\beta+1}} \tag{218}$$

Note that the positive term is linear in the sum of the $\rho_g$ whereas the negative term is sublinear. Now consider the marketplace's utility in the intervention scenario.

$$w^f(p) = \sum_{g \in G} p_g \sum_{i=1}^{N} \mathbf{1}[\mu_{i,g} \geq p_g] \mathcal{G}(y_g^{([M])}) \tag{219}$$

$$= \sum_{g \in G} \rho_g \left( Z - \alpha (y_g^{([M])})^{-\beta} \right) \tag{220}$$

$$= Z \sum_{g \in G} \rho_g - \alpha \sum_{g \in G} \rho_g (y_g^{([M])})^{-\beta} \tag{221}$$

$$= Z \sum_{g \in G} \rho_g - \alpha \sum_{g \in G} \rho_g \gamma_g^{-\beta} \left( \frac{\alpha \beta}{\kappa^T \gamma} \sum_{g \in G} \rho_g \gamma_g^{-\beta} \right)^{-\frac{\beta}{\beta+1}} \tag{222}$$

$$= Z \sum_{g \in G} \rho_g - \frac{\alpha}{M} \frac{\alpha \beta}{\kappa^T \gamma}^{-\frac{\beta}{\beta+1}} \left( \sum_{g \in G} \rho_g \gamma_g^{-\beta} \right)^{\frac{1}{\beta+1}} \tag{223}$$

Note that the positive term is linear in the sum of the $\rho_g$ whereas the negative term is sublinear. Consequently, the marketplace's utility ratio in the limit is

$$\lim_{N \to \infty} \frac{w^f(p)}{w(p)} = \lim_{N \to \infty} \frac{Z \sum_{g \in G} \rho_g}{Z \sum_{g \in G} \rho_g} = 1 \tag{224}$$

Consider seller $j$'s utility in the baseline scenario.

$$v_j(x^{(j)}) = \sum_{g \in G} p_g \sum_{i=1}^{N} \mathbf{1}[\mu_{i,g} \geq p_g] \sum_{T \subseteq [M] \setminus \{j\}} c_T \left( \mathcal{G}(x_g^{(T \cup \{j\})}) - \mathcal{G}(x_g^{(T)}) \right) - \sum_{g \in G} \kappa_g x_g^{(j)} \tag{225}$$

$$= \sum_{g \in G} \rho_g \frac{1}{M} \mathcal{G}(x_g^{([M])}) - \sum_{g \in G} \kappa_g x_g^{(j)} \tag{226}$$

$$= \sum_{g \in G} \rho_g \frac{1}{M} \left( Z - \alpha (x_g^{([M])})^{-\beta} \right) - \sum_{g \in G} \kappa_g x_g^{(j)} \tag{227}$$

$$= \frac{Z}{M} \sum_{g \in G} \rho_g - \frac{\alpha}{M} \sum_{g \in G} \rho_g \left( \frac{\rho_g}{\kappa_g} \alpha \beta \right)^{-\frac{\beta}{\beta+1}} - \sum_{g \in G} \kappa_g \frac{1}{M} \left( \frac{\rho_g}{\kappa_g} \alpha \beta \right)^{\frac{1}{\beta+1}} \tag{228}$$

$$= \frac{Z}{M} \sum_{g \in G} \rho_g - \frac{\alpha}{M} \sum_{g \in G} \rho_g^{\frac{1}{\beta+1}} \left( \frac{\alpha \beta}{\kappa_g} \right)^{-\frac{\beta}{\beta+1}} - \sum_{g \in G} \kappa_g \frac{1}{M} \left( \frac{\rho_g}{\kappa_g} \alpha \beta \right)^{\frac{1}{\beta+1}} \tag{229}$$

Note that the positive term is linear in the sum of the $\rho_g$ whereas the two negative terms are sublinear. Now consider seller $j$'s utility in the intervention scenario.

$$v_j^f(y^{(j)}) = \sum_{g \in G} p_g \sum_{i=1}^{N} \mathbf{1}[\mu_{i,g} \geq p_g] \sum_{T \subseteq [M] \setminus \{j\}} c_T \left( \mathcal{G}(y_g^{(T \cup \{j\})}) - \mathcal{G}(y_g^{(T)}) \right) - \sum_{g \in G} \kappa_g y_g^{(j)} \tag{230}$$

$$= \sum_{g \in G} \rho_g \frac{1}{M} \mathcal{G}(y_g^{([M])}) - \sum_{g \in G} \kappa_g y_g^{(j)} \tag{231}$$

$$= \sum_{g \in G} \rho_g \frac{1}{M} \left( Z - \alpha (y_g^{([M])})^{-\beta} \right) - \sum_{g \in G} \kappa_g y_g^{(j)} \tag{232}$$

$$= \frac{Z}{M} \sum_{g \in G} \rho_g - \frac{\alpha}{M} \sum_{g \in G} \rho_g \gamma_g^{-\beta} \left( \frac{\alpha \beta}{\kappa^T \gamma} \sum_{g \in G} \rho_g \gamma_g^{-\beta} \right)^{-\frac{\beta}{\beta+1}} - \sum_{g \in G} \frac{\kappa_g \gamma_g}{M} \left( \frac{\alpha \beta}{\kappa^T \gamma} \sum_{g \in G} \rho_g \gamma_g^{-\beta} \right)^{\frac{1}{\beta+1}} \tag{233}$$

$$= \frac{Z}{M} \sum_{g \in G} \rho_g - \frac{\alpha}{M} \frac{\alpha \beta}{\kappa^T \gamma}^{-\frac{\beta}{\beta+1}} \left( \sum_{g \in G} \rho_g \gamma_g^{-\beta} \right)^{\frac{1}{\beta+1}} - \sum_{g \in G} \frac{\kappa_g \gamma_g}{M} \left( \frac{\alpha \beta}{\kappa^T \gamma} \sum_{g \in G} \rho_g \gamma_g^{-\beta} \right)^{\frac{1}{\beta+1}} \tag{234}$$

Note that the positive term is linear in the sum of the $\rho_g$ where as the two negative terms are sublinear. Consequently, seller $j$'s utility ratio in the limit is

$$\lim_{N \to \infty} \frac{v_j^f(y^{(j)})}{v_j(x^{(j)})} = \frac{\frac{Z}{M} \sum_{g \in G} \rho_g}{\frac{Z}{M} \sum_{g \in G} \rho_g} = 1. \tag{235}$$

Consider buyer $i$'s utility in the baseline scenario,

$$u_i(\mu_i) = \sum_{g \in G} (\mu_{i,g} - p_g) \mathcal{G}(\mathcal{AF}_g(\mu_{i,g}, x^{([M])})). \quad (236)$$

Buyer $i$ will be allocated data for group $g$ if and only if $\mu_{i,g} \geq p_g$. Let $G_p \triangleq \{g \in G : \mu_{i,g} \geq p_g\}$, and consider the buyer's utility in the limit,

$$\lim_{N \to \infty} u_i(\mu_i) = \lim_{N \to \infty} \sum_{g \in G_p} (\mu_{i,g} - p_g) \mathcal{G}(x_g^{([M])}) \leq Z \sum_{g \in G_p} (\mu_{i,g} - p_g), \quad (237)$$

where the inequality follows because for all $g$ and $x_g^{([M])}$, $\mathcal{G}(x_g^{([M])}) \leq Z$. By assumption, there is at least one group such that $\max_{p_g} \rho_g \to \infty$ as $N \to \infty$. For any such group it follows that $\mathcal{G}(x_g^{([M])}) \to Z$ as $N \to \infty$ because $x_g^{([M])} \to \infty$ by Claim (A.2).

Now consider buyer $i$'s utility in the intervention scenario.

$$u_i^f(\mu_i) = \sum_{g \in G} (\mu_{i,g} - p_g) \mathcal{G}(\mathcal{AF}_g(\mu_{i,g}, y^{([M])})). \quad (238)$$

Again, buyer $i$ will be allocated data for group $g$ if and only if $\mu_{i,g} \geq p_g$. Therefore, the buyer's utility in the limit is

$$\lim_{N \to \infty} u_i^f(\mu_i) = \lim_{N \to \infty} \sum_{g \in G_p} (\mu_{i,g} - p_g) \mathcal{G}(y_g^{([M])}) = Z \sum_{g \in G_p} (\mu_{i,g} - p_g). \quad (239)$$

The last equality follows from the assumption that there is at least one group such that $\max_{p_g} \rho_g \to \infty$ as $N \to \infty$. By Claim (A.3), it follows that for every group $g$, $y_g^{([M])} \to \infty$ and consequently $\mathcal{G}(y_g^{([M])}) \to Z$.

Putting these together we can evaluate the cost of fairness in the limit

$$\lim_{N \to \infty} \frac{u_i^f(\mu_i)}{u_i(\mu_i)} \geq \frac{Z \sum_{g \in G_p} (\mu_{i,g} - p_g)}{Z \sum_{g \in G_p} (\mu_{i,g} - p_g)} = 1. \quad (240)$$

$\square$

