# OpenReview forum: "The Cost of Balanced Training-Data Production in an Online Data Market"
_ACM.org/TheWebConf/2025/Conference — WWW 2025 Poster_

### Official Review · Reviewer_uuMX · 2024-11-10

**Novelty:** 5
**Technical Quality:** 5

**Review:**

This paper focused on the online data market, where the authors intended to address the ethical issues and achieve balanced training-data production. After modeling, the paper proposed two main insights: (1) In small and emerging markets, an intervention can drive the data producers out of the market, so that the cost of fairness is maximal; (2) in large and established markets, the cost of fairness can vanish (as a fraction of overall welfare) as the market grows.

Pros:

This paper had complete model regarding the data market with sellers, buyers, and a centralized market place. The authors proposed detailed proofs for the propositions, facts, lemmas, and claims.

Cons:

I have three major concerns after reading this paper: (1) The function of the centralized marketplace is significantly powerful, and it has too much information. Thus, what will happen if the marketplace behaves dishonestly? (2) This paper studied the problem of unbalanced training data as an example, while it is too simple to represent the complex ethical issues. (3) Simulations can be conducted to provide an intuitive understanding of the insights. The details will be discussed in Questions.

**Questions:**

At first, I think the function of the centralized marketplace is significantly powerful. It seems that the centralized marketplace can almost fully obtain the information of sellers. In this case, the centralized marketplace may have opportunity to control the trading. Therefore, what will happen if the centralized marketplace behaves dishonestly? Is there any mechanism to regulate the centralized marketplace?

Second, this paper studied the problem of unbalanced training data as an example, while it is too simple to represent the complex ethical issues as raised by the authors, such as “Models can produce predictions that systematically differ by race [40], reproduce the photos of individuals in their training data [11], be trained on copyrighted materials [37], and require the labeling of psychologically harmful content [44].” This concern was also mentioned in Limitations and Discussion, while I think it is very important to check whether the proposed framework can fit the more complex ethical issues.

Third, there are no simulations regarding the game. Numerical simulations with corresponding figures can provide intuitive insights for readers to better understand the results.

Some points can be improved:

1. In the Introduction, I think it is much better to provide a practical example to further clarify the application scenario regarding the content “Recently, a small and growing number of firms have been attempting to address ethical issues in training data as profit-seeking participants in online data markets. They are targeting a range of issues including protecting creators’ copyright [50], creating job opportunities with fair working conditions [29], and producing representative training data [18]. And they are working together [4]. This trend is intriguing.”

2. It is much better to provide a figure in Section 3.1 to clarify the entities and procedure of the game.

3. Typos and grammar issues can be further improved, e.g., “A dataset could could be comprised of images of people”, “Learning curve A learning curve”.

4. The proposed example in Definition 3.3 seems not like a case of marginal value-of-accuracy “For example, an ecommerce company may estimate that every 1% of accuracy in transcribing spoken words yields revenues of $10,000. If x_g training samples results in overall accuracy of 73%, the company would expect revenues of $730,000 and be willing to pay up to that amount for those predictions.”

5. It is better to provide an intuitive explanation to readers regarding the Z, alpha, beta in Definition 3.2.

6. It is better to provide some reference for “Shapley value”, since not all readers are familiar with the concept.

7. The format of formulas (116) and (117) are strange.

**Reviewer Confidence:**

3: The reviewer is confident but not certain that the evaluation is correct

**Scope:**

3: The work is somewhat relevant to the Web and to the track, and is of narrow interest to a sub-community

---

### Official Review · Reviewer_sVL6 · 2024-11-28

**Novelty:** 2
**Technical Quality:** 2

**Review:**

Strengths
1.The authors have developed a sophisticated model that captures the complexities of online data markets, including the dynamics between data sellers, buyers, and the marketplace.
2.The paper offers valuable insights into how market conditions, such as size and maturity, can influence the cost of fairness.
Weaknesses
1.The paper lacks illustrative diagrams to effectively clarify the modeling process of the online market model.
2.The manuscript presents a theoretical model without empirical validation. Incorporating real-world data to test the model's predictions could strengthen the conclusions.
3.The study focuses on a single criterion of fairness related to dataset demographics. A broader analysis that includes multiple fairness criteria could provide a more comprehensive understanding of the costs and benefits of ethical interventions.

**Questions:**

1.Refer to weaknesses.
2.How generalizable are the findings to different types of online data markets, especially those with varying structures and incentive mechanisms?

**Reviewer Confidence:**

3: The reviewer is confident but not certain that the evaluation is correct

**Scope:**

3: The work is somewhat relevant to the Web and to the track, and is of narrow interest to a sub-community

---

### Official Review · Reviewer_fCQT · 2024-11-29

**Novelty:** 6
**Technical Quality:** 6

**Review:**

Pros:

- Robust theoretical analysis
- Nice trending topic and conclusions
- Well-written

Cons:

- Some assumptions and limitations may reduce the scope of application of the theoretical results
- Limited practical application

The paper touches upon the trending topic of fairness and ethical AI/ML, combined with data marketplaces (DMs). In the context of a specific theoretical model of a data marketplace (Agarwal et al. [2]), it calculates the equilibrium of a free market where sellers are free to produce samples of different groups, even if they create an unbalanced training set (summing the contributions of all sellers) for the market to train a model and calculate predictions for buyers. Then, this result is compared to the equilibrium when the market intervenes to shape the balance of training samples to fit a certain target distribution the market considers as *balanced*.

I find the topic and the rationale of the paper very interesting. I am now aware of any paper that touches upon the topic of DMs intervening to ensure a balanced distribution of a training set. The paper is mainly theoretical, although it manages to distill the results and present them in an attractive way to the reader. The results are interesting, and it predicts the feasibility of balanced data markets in the setting and conditions set by the author(s).

The paper organization is fine and it is well written. However, it is a complex paper sometimes difficult to follow due to the notation. I think it could benefit from a table summarising the notation, avoiding to go back and forth to search for the meaning of variables.

In Sect. 2 related literature, I miss papers related to data selection mechanisms in DMs that allow buyers to appraise data according to a goal set by sellers before buying or using them. An alternative to DMs intervening to achieve a balanced training set is giving buyers the possibility to do so by tailoring their data selection and including notions of fairness in their goal, which these methodologies allow for. Some papers on the topic are the following:

* **[Data Station: Delegated, Trustworthy, and Auditable Computation to Enable Data-Sharing Consortia with a Data Escrow.](https://raulcastrofernandez.com/papers/data_station_paper-11.pdf)** Siyuan Xia, Zhiru Zhu, Chris Zhu, Jinjin Zhao, Kyle Chard, Aaron Elmore, lan Foster, Michael Franklin, Sanjay Krishnan, Raul Castro Fernandez. **VLDB 2022**
* Santiago Andrés Azcoitia and Nikolaos Laoutaris. 2022. Try before you buy: a practical data purchasing algorithm for real-world data marketplaces. In Proceedings of the 1st International Workshop on Data Economy (DE '22). ACM
* Data Appraisal Without Data Sharing. Xinlei Xu, Awni Hannun, Laurens Van Der Maaten *Proceedings of The 25th International Conference on Artificial Intelligence and Statistics* , PMLR 151:11422-11437, 2022.

There are some assumptions that can be justififed better, even when they are common in related literature:

1. Value extracted separately among groups by buyers. How does this serve the purpose of general models / predictions applicable to different population groups? I think in many real use cases there exists transfer of knowledge between groups.
2. Marginal value of accuracy is constant. This does not happen in practice, where the marginal value is zero up to a certain point, and not necessarily linear from that point on. Instead of explaining how the linear hypothesis work, I miss a justification about why this is representative and an intuition on the learning curve and the role of their parameters (or alternatively a more precise reference to where this function is explained).

My main concern is with the limitations and practical application of the paper. Maybe the authors(s) can emphasise or extend on it.

Regarding the DM model, the paper focuses on the one defined in reference [2]. I am not aware of any practical implementation of this model that has shown it is feasible in practice. Will buyers be willing to give out the task (including a pre-trained model) + (I assume, the paper does not mention it) input features for new predictions to be given back just predictions of a new model including the training data of the DM model? How does this model compare for example to federated learning or model-based marketplaces where buyers get back the model for them to exploit?

Finally, the paper focuses on one specific fairness criteria, and the authors expect that economic growth can dampen (typo in the paper in this sentence) the cost of fairness for other criteria, as well. Is there any justification for this statement?

Regarding references, some of them point to preprints and must refer to the conference or journal paper. Some examples: ref 16 is published in the Journal of LM Research, ref 20 is in PMLR, ref 27 is NIPS, ref 31 is published in the ITCS conference.

Overall, I liked and find the paper a very valuable theoretical work that leads to powerfull conclusions, but I think it can improve by connecting this theoretical exercise with real-world data marketplaces close to (or that can implement) this model.

**Questions:**

Can the author(s) provide an intuition to the learning curve and the role of the parameters there? Or point to a specific reference or section that uses and describes the learning curve with this exact notation.

Are the author(s) aware of any practical implementation or experiment following the DM model defined by reference [2]? If so, referring to it may increase the credibility of the paper.

Can the author(s) elaborate on why they believe the cost of fairness will amortize with market size for any fairness criteria?

**Reviewer Confidence:**

3: The reviewer is confident but not certain that the evaluation is correct

**Scope:**

4: The work is relevant to the Web and to the track, and is of broad interest to the community

---

### Official Review · Reviewer_G36R · 2024-11-30

**Novelty:** 5
**Technical Quality:** 5

**Review:**

This paper tries to define an online market that sustainably and efficiently addresses ethical issues in the machine-learning economy, in which there is a set of sellers and buyers. These sellers produce and sell data to the marketplace. These buyers want to buy predictions from the marketplace. The centralized marketplace coordinates the market by aggregating and allocating date, producing predictions by carrying out machine learning, and setting prices.


I have to say that I am not the right person to judge the technical novelty of this paper. I am usually working on approximation and online algorithms. To me, this paper studied an interesting problem that was also motivating. Moreover, I don't see any previous work that aims to theoretically define an ethical data market. This work can be considered as the first such work.

**Questions:**

I don't have any specific questions.

**Reviewer Confidence:**

2: The reviewer is willing to defend the evaluation, but it is likely that the reviewer did not understand parts of the paper

**Scope:**

4: The work is relevant to the Web and to the track, and is of broad interest to the community

---

### Official Review · Reviewer_rmCt · 2024-11-30

**Novelty:** 4
**Technical Quality:** 5

**Review:**

The paper addresses an important topic in the economics of online data markets, examining the cost of achieving fairness in data production and its implications for market outcomes. By focusing on a stylized model, the authors explore the dynamics of fairness interventions, their feasibility under varying market conditions, and their implications for market participants.

The relevance of the topic cannot be overstated, given the increasing role of data in economic transactions and the growing emphasis on ethical considerations in machine learning. However, several aspects of the paper raise questions about the assumptions, implications, and novelty of the findings.

I am not a theorist so some of my concerns may reflect a lack of familiarity with the nuances of theoretical modeling. However, I believe some questions are important to ensure the findings are both robust and relevant.

First, the key results—that fairness constraints can deter market participation when costs are high, but increased demand can offset fixed costs and enable market formation—are rather intuitive to me. The paper could benefit from addressing why a formal model is necessary to derive these conclusions and what specific insights the model offers beyond these predictable outcomes. For example, what makes data markets distinct from other types of markets where similar dynamics might apply? This clarification would strengthen the paper's contribution to the broader literature.

Second, the paper assumes that groups are independent in their data characteristics, a simplification that does not align with many real-world scenarios. For example, binary variables such as gender inherently imply correlations across groups. This independence assumption limits the model's applicability to cases where such correlations matter. While incorporating group correlations would likely make the mathematics more complex and less tractable, it would be useful for the authors to discuss how correlated group structures might alter the results.

The third point is my biggest concern. The operationalization of fairness in the model relies on the marketplace's ability to judge whether datasets meet demographic balance criteria. The marketplace would need to know the ground truth to do that, no? Wouldn’t it be better to model fairness very differently, on the buyer side, where buyers only buy data from one supplier if it sufficiently increases their prediction accuracy over data offered by another supplier? In this way, no one needs to know the ground truth. Side note. Focusing on prediction accuracy more explicitly would also introduce the notion of variance in the output of the learning on the buyer side, which is something I (as an outsider to this literature!) miss in the discussion about model fairness / ethical AI. Wouldn't perceived bias be much less of a problem if AI applications would not take point estimates and portray them as having a zero standard error, but openly communicate something like a confidence band?

Finally, the paper does not sufficiently explore the welfare and policy implications of its findings. While the authors note that fairness interventions can impose costs on market participants, they do not examine how these costs might translate into broader societal benefits, such as improved outcomes for end users or increased trust in data markets. A welfare analysis incorporating these factors could provide a more balanced view of the trade-offs involved. Moreover, the role of public policy in addressing fairness, such as through subsidies for data production, is largely absent. For instance, government-funded census data illustrates how public intervention can address issues of representativeness in contexts where private incentives seem insufficient.

By addressing or at least discussing these points, the authors could enhance the impact of their work and provide a more comprehensive framework for understanding the economics of fairness in online data markets.

**Questions:**

1. What specific insights or mechanisms in your model make data markets distinct from other markets where similar dynamics might apply?

2. How would your conclusions change if group characteristics were correlated rather than independent? For example, how might modeling binary variables like gender, where membership in one group implies exclusion from another, alter your results?

3. Your model assumes that the marketplace can evaluate whether datasets meet demographic balance criteria. What mechanism or data would the marketplace realistically use to assess fairness without access to the ground truth distribution? Have you considered alternative fairness criteria that do not rely on ground truth knowledge?

4. Would you consider shifting the fairness criterion to the buyer side, where buyers only purchase data that improves their prediction accuracy relative to other suppliers? How might this alternative framework impact your model’s conclusions?

5. Your analysis highlights the costs of fairness interventions but does not explore their broader societal benefits, such as improved end-user utility or trust in data markets. Have you considered the role of public policy, such as subsidies for data production, in mitigating these costs and fostering fairness?

**Ethics Review Flag:**

Yes

**Reviewer Confidence:**

2: The reviewer is willing to defend the evaluation, but it is likely that the reviewer did not understand parts of the paper

**Scope:**

3: The work is somewhat relevant to the Web and to the track, and is of narrow interest to a sub-community